METHODS AND RESOURCES

# A versatile new tool derived from a bacterial deubiquitylase to detect and purify ubiquitylated substrates and their interacting proteins

Mengwen Zhang[1], Jason M. Berk[2¤], Adrian B. Mehrtash[3], Jean Kanyo[4], Mark Hochstrasser[2,3]*

1 Department of Chemistry, Yale University, New Haven, Connecticut, United States of America,
2 Department of Molecular Biophysics and Biochemistry, Yale University, New Haven, Connecticut, United States of America, 3 Department of Molecular, Cellular, and Developmental Biology, Yale University, New Haven, Connecticut, United States of America, 4 W.M. Keck Foundation Biotechnology Resource Laboratory, Yale University, New Haven, Connecticut, United States of America

¤ Current address: Arvinas, Inc., New Haven, Connecticut, United States of America
* mark.hochstrasser@yale.edu

**Data Availability Statement:** All relevant data are within the paper and its Supporting Information files. All mass spectrometry proteomics data obtained in this study have been deposited to the

## Abstract

Protein ubiquitylation is an important posttranslational modification affecting a wide range of cellular processes. Due to the low abundance of ubiquitylated species in biological samples, considerable effort has been spent on methods to purify and detect ubiquitylated proteins. We have developed and characterized a novel tool for ubiquitin detection and purification based on OtUBD, a high-affinity ubiquitin-binding domain (UBD) derived from an *Orientia tsutsugamushi* deubiquitylase (DUB). We demonstrate that OtUBD can be used to purify both monoubiquitylated and polyubiquitylated substrates from yeast and human tissue culture samples and compare their performance with existing methods. Importantly, we found conditions for either selective purification of covalently ubiquitylated proteins or co-isolation of both ubiquitylated proteins and their interacting proteins. As proof of principle for these newly developed methods, we profiled the ubiquitylome and ubiquitin-associated proteome of the budding yeast *Saccharomyces cerevisiae*. Combining OtUBD affinity purification with quantitative proteomics, we identified potential substrates for the E3 ligases Bre1 and Pib1. OtUBD provides a versatile, efficient, and economical tool for ubiquitin research with specific advantages over certain other methods, such as in efficiently detecting monoubiquitylation or ubiquitin linkages to noncanonical sites.

## Introduction

Ubiquitin is a conserved posttranslational modifier that requires a cascade of enzymatic reactions for its attachment to proteins [1]. Each modification is catalyzed by a ubiquitin-activating enzyme E1, ubiquitin-conjugating enzyme E2, and ubiquitin ligase E3 [2]. The E1 enzyme activates the carboxyl-terminal carboxylate of ubiquitin and then transfers the activated ubiquitin

ProteomeXchange Consortium via the PRIDE partner repository with the dataset identifier PXD032294 (yeast results) and PXD032675 (HeLa cell results). The raw and fully uncropped images for all gels and blots have been deposited to Figshare (https://figshare.com/articles/figure/S1_raw_images_pdf/19640367).

**Funding:** This work was supported by National Institutes of Health (https://www.nih.gov/grants-funding) grant GM136325 to M.H. The mass spectrometers at the Keck MS & Proteomics Resource at Yale University were funded in part by the Yale School of Medicine and by the Office of the Director, National Institutes of Health (S10OD02365101A1, S10OD019967, and S10OD018034). The funders had no role in study design, data collection and analysis, decision to publish, or preparation of the manuscript.

**Competing interests:** I have read the journal's policy and the authors of this manuscript have the following competing interests: M.H., M.Z. and J.M. B. are inventors on a U.S. Patent Application No. No. 17/061,347 filed on October 1, 2020 that covers methods of ubiquitin detection and enrichment using the OtUBD. The other authors declare no competing interests.

**Abbreviations:** ACA, aminocaproic acid; BSA, bovine serum albumin; DBPS, Dulbecco's phosphate buffered saline; DMEM, Dulbecco's Modified Eagle Medium; DMP, dimethyl pimelimidate; DTT, dithiothreitol; DUB, deubiquitylase; FPLC, Fast Protein Liquid Chromatography; GG, GlyGly; IAA, iodoacetamide; IMAC, immobilized metal affinity chromatography; IPTG, isopropyl β-D-1-thiogalactopyranoside; LB, Luria-Bertani; LC-MS/MS, liquid chromatography with tandem mass spectrometry; MBP, maltose-binding protein; MMTS, methyl methanethiosulfonate; NEM, N-ethylmaleimide; PMSF, phenylmethylsulfonyl fluoride; RNAPII, RNA polymerase II; TCEP, tris(2-carboxyethyl) phosphine); TUBE, tandem ubiquitin-binding entity; UBD, ubiquitin-binding domain; WT, wild-type; 4-NQO, 4-nitroquinoline-1-oxide.

molecule to an E2. E3 ligases are responsible for the recognition of the substrate and catalyzing ubiquitin transfer from the E2 to a nucleophilic residue on the substrate protein, typically the ε-amino group of a lysine residue, but potentially also N-terminal amino groups, serine/threonine hydroxyl side chains, or the thiol group of cysteine [3]. Ubiquitin itself can be ubiquitylated through its N-terminal methionine (M1) or one or more of its 7 lysine residues (K6, K11, K27, K29, K33, K48, and K63) [4]. These diverse ubiquitin chain topologies and sizes can modulate the biological functions of substrate ubiquitylation, often described as the "ubiquitin code" [5]. For example, monoubiquitylation has been reported to facilitate protein complex formation in many cases [6,7]. Polyubiquitylation involving K48 linkages is a well-documented substrate marker for proteasomal degradation [8], while polyubiquitylation with K63 linkages is often a signal for membrane trafficking or DNA repair pathways [9,10]. Ubiquitylation can be reversed through hydrolysis by ubiquitin-specific proteases or deubiquitylases (DUBs) [11].

Defects in ubiquitylation have been connected to many human disorders, including cancers, viral infections, and neurodegenerative diseases [12–14]. The broad biomedical impact of protein ubiquitylation has stimulated efforts to develop sensitive methods to study the ubiquitylated proteome [15,16]. Because the ubiquitylated fraction of a given protein substrate population is often very small at steady state [17], it is generally necessary to enrich for the ubiquitylated proteins in biological samples of interest. Current methods to enrich ubiquitylated proteins can be roughly classified into 3 categories: (1) ectopic (over)expression of epitope-tagged ubiquitin and affinity purification using the tags; (2) immunoprecipitation with anti-ubiquitin antibodies; and (3) use of tandem ubiquitin-binding entities (TUBEs) [18–20,21–24].

The first method was introduced using the budding yeast *Saccharomyces cerevisiae* [25]. In yeast, 4 different genes encode ubiquitin, either as fusions to ribosomal peptides or as tandem ubiquitin repeats [26]. It is possible to create yeast strains where the only source of ubiquitin is a plasmid expressing epitope-tagged ubiquitin [27,28]; as a result, all ubiquitylated proteins in this specific yeast strain bear the epitope tag, which can then be used for enrichment or detection of the ubiquitylated species. A number of earlier studies have used this method to profile the ubiquitylated proteome [4]. One major concern with this method is that the (over)expression of tagged ubiquitin may result in abnormal ubiquitylation or interfere with endogenous ubiquitylation events.

To study endogenous ubiquitylated proteins, anti-ubiquitin antibodies—including those against all ubiquitylation types (such as FK1 and FK2 monoclonal antibodies [29]) or those specific for certain ubiquitin-chain linkages (such as anti-K48 ubiquitin linkage antibodies)—have been used [30,31]. TUBEs, on the other hand, are recombinant ubiquitin-affinity reagents built from multiple ubiquitin-binding domains (UBDs). UBDs have been characterized in a range of ubiquitin-interacting proteins, and they typically bind to ubiquitin with low affinity (Kd values in the micromolar range) [32]. By fusing multiple copies of a UBD together to turn it into a TUBE, the avidity of the reagent toward polyubiquitin chain-modified proteins is greatly increased [21]. TUBEs are therefore useful in protecting polyubiquitylated proteins from DUB cleavages and enriching them in biological samples, and some TUBEs are designed to recognize specific types of polyubiquitin chains [33]. In general, TUBE affinity toward monoubiquitylated proteins is low [21].

In addition to the abovementioned methods, ubiquitin remnant motif antibodies (diGly antibodies) are widely used in bottom-up proteomics experiments to identify ubiquitylation sites on substrate proteins [34,35]. In bottom-up proteomics, proteins are digested by a protease (typically trypsin) into short peptides, separated by liquid chromatography and identified by tandem mass spectrometry (LC-MS/MS) [36]. Tryptic digestion of ubiquitylated proteins leaves a signature GlyGly (GG) remnant on ubiquitylated lysine side chains [18]. Anti-diGly-

ε-Lys antibodies recognize this remnant motif and enrich such peptides for identification of ubiquitylation sites. The development of diGly antibodies has greatly facilitated the systematic discovery and profiling of ubiquitylated proteins and their ubiquitylation sites and has enabled the establishment of databases documenting ubiquitylation in humans and other species [37,38].

Each of these methods has its advantages and limitations, which have been reviewed elsewhere [16,39]. For example, TUBEs are excellent tools to study polyubiquitylation, but in some mammalian cell types, over 50% of ubiquitylated proteins are only monoubiquitylated [17] and can easily be missed by TUBEs. Anti-diGly antibodies, while extremely effective in identifying ubiquitin–lysine linkages, are not capable of recognizing ubiquitylation sites on other nucleophilic side chains in proteins or other macromolecules [40]. Due to the importance and complexity of ubiquitylation, the development of sensitive and economical reagents to study the entire ubiquitylome is crucial.

Recently, our group discovered a novel UBD within a DUB effector protein, OtDUB, from the intracellular bacterium *Orientia tsutsugamushi*, the causative agent of the disease scrub typhus [41]. The UBD from OtDUB, which was referred to as OtDUB$_{UBD}$ (we will use OtUBD for the remainder of the paper for simplicity), spans residues 170 to 264 of the 1,369-residue OtDUB polypeptide (Fig 1A) and binds monomeric ubiquitin with very high affinity (K$_d$, approximately 5 nM), which is more than 500-fold tighter than any other natural UBD described to date. Co-crystal structures of OtDUB and ubiquitin revealed that OtUBD binds ubiquitin at the isoleucine-44 hydrophobic patch, a ubiquitin feature commonly recognized by ubiquitin-binding proteins [42]. We reasoned that the small, well-folded OtUBD could serve as a facile enrichment reagent for ubiquitylated proteins. The advantages of such reagent include its low cost, lack of bias between monoubiquitylated and polyubiquitylated proteins, and ability to detect unconventional ubiquitin-substrate linkages.

## Results

### OtUBD can protect and enrich ubiquitylated species from whole cell lysates

We first expressed and purified recombinant OtUBD with an N-terminal His$_6$ tag (Fig 1B). A previously reported TUBE based on the UBA domain of human Ubiquilin 1 (4xTR-TUBE) was used for comparison [23,43]. One use of TUBEs is to protect ubiquitylated proteins in vitro from being cleaved by endogenous DUBs or being degraded by the proteasome following cell lysis, which facilitates their analysis [21]. We tested if OtUBD could do the same. When yeast cells were lysed in the presence of N-ethylmaleimide (NEM; a covalent cysteine modifier that inhibits most cellular DUBs), 3 μM OtUBD, or 3 μM TUBE, higher mass ubiquitylated species were similarly preserved by the 2 ubiquitin binders, with NEM having the strongest effect, as expected (Fig 1C).

We investigated whether this protection extended to monoubiquitylated proteins by examining Flag-tagged histone H2B (Htb2) in a *ubp8Δ* mutant [44]. Histone H2B is known to be monoubiquitylated, and levels of this species are enhanced by deleting Ubp8, the DUB that reverses the modification [45]. Strikingly, OtUBD added to the cell lysate preserved the monoubiquitylated H2B to a degree comparable to NEM (Fig 1C, bottom). By contrast, H2B-ubiquitin was completely lost in extracts without any DUB inhibitor or when incubated with the TUBE protein.

We next determined if OtUBD or tandem repeats of OtUBD could be used for affinity enrichment of ubiquitylated proteins. We fused maltose-binding protein (MBP) to the amino terminus of OtUBD or 3 tandem OtUBD repeats (Fig 1B). Purified MBP or the MBP fusion proteins were first bound to an amylose resin and then incubated with yeast whole cell lysates.

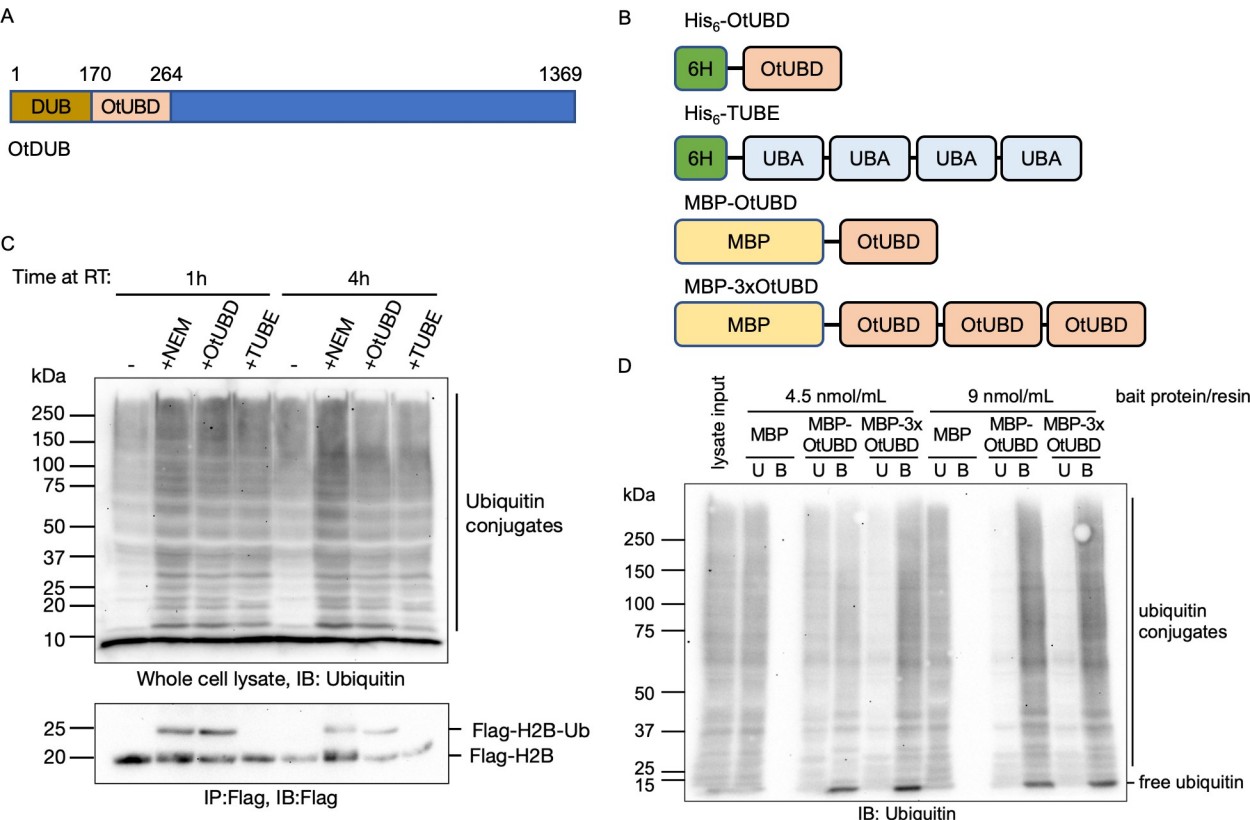

**Fig 1. The high-affinity UBD from OtDUB efficiently protects and enriches for yeast ubiquitylated species. (A)** Schematic showing the ubiquitin binding domain (OtUBD) within the *O. tsutsugamushi* DUB (OtDUB). OtUBD spans residues 170 to 264. **(B)** The different constructs of OtUBD and the control TUBE derived from the UBA domain of human Ubiquilin 1. His$_6$ tagged OtUBD and TUBE were used in the ubiquitylation protection experiment shown in Fig 1C and MBP-tagged OtUBD and 3xOtUBD were used in the ubiquitin pulldown experiment in Fig 1D. **(C)** IB analysis of bulk ubiquitylated proteins (top panel) and histone H2B (bottom panel) from yeast cell lysates prepared in the presence of different reagents. OtUBD prevents deubiquitylation of bulk ubiquitylated substrates (top panel) and monoubiquitylated histone H2B (bottom panel). **(D)** IB analysis of MBP pulldowns from yeast cell lysates using different bait proteins. MBP or MBP-tagged bait proteins bound to an amylose resin were incubated with yeast lysates, and bound proteins were eluted by incubation in SDS sample buffer. Both OtUBD and 3xOtUBD bound (B) ubiquitylated substrates in the lysates. Concentration of bait protein indicates the amount of bait protein per unit volume of amylose resin. B, bound fraction; IB, immunoblot; MBP, maltose-binding protein; OtDUB, *O. tsutsugamushi* DUB; TUBE, tandem ubiquitin-binding entity; U, unbound fraction; UBD, ubiquitin-binding domain.

With lower amounts of the resin-bound bait proteins, MBP-3xOtUBD enriched more ubiquitylated proteins than MBP-OtUBD, likely due to its higher capacity for binding ubiquitin (3 ubiquitin binding sites versus 1 in OtUBD) (Fig 1D, left; compare bound (B) to unbound (U) lanes). When we increased the amount of the bait proteins, however, both MBP-OtUBD and MBP-3xOtUBD efficiently depleted ubiquitylated proteins from the lysate (Fig 1D, right). The negative control MBP did not detectably bind any ubiquitylated species at either concentration. Notably, efficient enrichment was only achieved when MBP-OtUBD was prebound to the amylose resin (S1A Fig). When free MBP-OtUBD was first incubated with the cell lysate and then bound to amylose resin, the enrichment efficiency was compromised (S1B Fig). MBP-OtUBD also efficiently enriched ubiquitylated proteins from mammalian cell lysates, demonstrating its general utility across species (S1C Fig).

In summary, OtUBD can both protect ubiquitylated proteins from in vitro deubiquitylation and enrich for such proteins. Unlike previously reported UBDs [21,46], OtUBD can efficiently enrich ubiquitylated proteins even when used as a single entity instead of tandem repeats.

## A covalently linked OtUBD resin for ubiquitylated protein purification

We next generated resins with covalently attached OtUBD to minimize the contamination by bait proteins seen with MBP-OtUBD and maltose elution (S1A–S1C Fig). Since OtUBD lacks cysteine residues, we introduced a cysteine residue at the amino terminus of the OtUBD sequence as a functional handle that can react with the commercially available SulfoLink resin to form a stable thioether linkage (Fig 2A). As a negative control, free cysteine was added to the SulfoLink resin to cap the reactive iodoacetyl groups. When incubated with yeast whole cell lysates prepared in a buffer with 300 mM NaCl and 0.5% Triton-X100 detergent, the OtUBD resin bound a broad range of ubiquitylated proteins and the bound proteins could be eluted with a low pH buffer (Figs 2B and S1D; see Materials and methods). No ubiquitylated species were detected in the eluates from the control resin (Figs 2B and S1D).

By comparing the anti-ubiquitin blot in Fig 2B to the general protein stain of the same eluted fractions in S1D Fig, it was clear that many proteins eluted from the OtUBD resin were not themselves ubiquitylated. Pulldown experiments performed under native or near-native

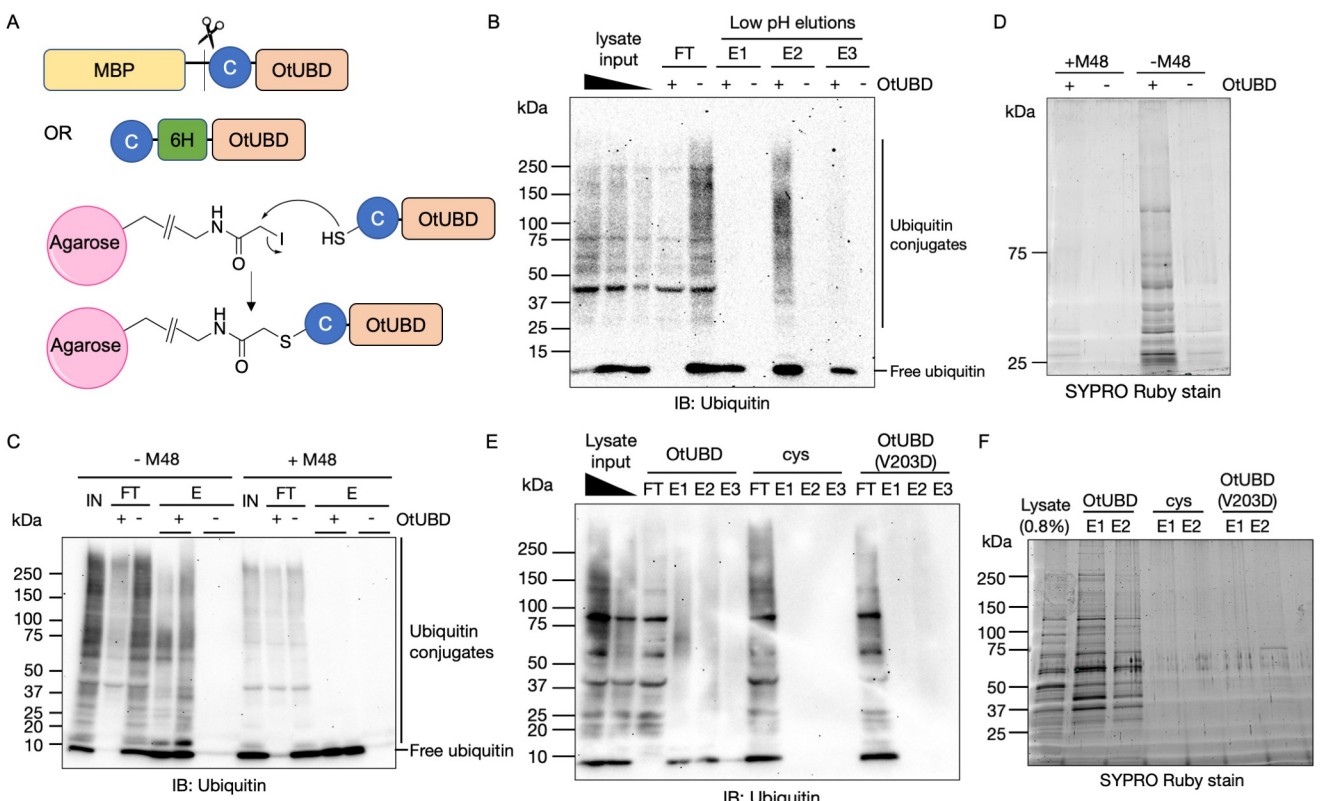

**Fig 2. A covalently linked OtUBD resin purifies ubiquitin and ubiquitylated proteins from yeast lysates. (A)** OtUBD constructs used for covalent coupling to resin and mechanism of the coupling reaction. An engineered cysteine at the amino terminus of OtUBD enables its covalent conjugation to the SulfoLink resin. **(B)** Ubiquitin blot of pulldowns from yeast cell lysate using covalently linked OtUBD resin or control resin. Covalently linked OtUBD resin efficiently pull down ubiquitylated species from yeast whole cell lysate. FT, flow-through; E1/E2/E3, eluted fractions using a series of stepwise, low pH elutions. **(C, D)** Extract pretreatment with M48 DUB cleaves ubiquitin from ubiquitylated species and greatly reduces the total protein pulled down by OtUBD resin. (C) Anti-ubiquitin blot of OtUBD pulldown of yeast lysate with or without M48 DUB treatment. (D) Total protein present in the eluted fractions of the OtUBD pulldowns visualized with SYPRO Ruby stain. (C and D are from 2 separate biological replicates.) IN, input; FT, flow-through; E, eluted fractions. **(E, F)** The V203D mutation in OtUBD, which greatly impairs its binding of ubiquitin, prevents enrichment for ubiquitylated species from yeast lysate. (E) Anti-ubiquitin blot of pulldowns of yeast lysates using OtUBD resin, Cys resin (negative control), and OtUBD(V203D) resin. (F) Total protein present in the eluted fractions of the OtUBD pulldowns visualized with SYPRO Ruby stain. IN, input; FT, flow-through; E1/2/3, eluted fractions using a series of low pH elutions. MBP, maltose-binding protein.

conditions are expected to co-purify proteins that interact noncovalently with ubiquitylated polypeptides, e.g., complexes that harbor ubiquitylated subunits. To test whether the entire protein population eluted from OtUBD resin was nevertheless dependent on substrate ubiquitylation, yeast lysates were preincubated with the viral M48 DUB, which cleaves a broad range of ubiquitylated proteins and reduces ubiquitin chains to free ubiquitin (Fig 2C) [47]. This treatment greatly reduced the total protein eluted from the OtUBD resin compared to the pull-down from untreated lysate (Fig 2D), indicating that the majority of proteins eluted from OtUBD resin were either ubiquitylated themselves or interacted noncovalently with ubiquitin or ubiquitylated proteins.

To further validate the specificity of the OtUBD resin toward ubiquitylated proteins, we made an OtUBD resin with a ubiquitin-binding deficient mutation (V203D) [41] and tested its ability to purify ubiquitin and ubiquitylated proteins. This mutation greatly diminished the resin's ability to enrich ubiquitylated species (Fig 2E) and also strongly reduced the total protein eluate from the resin (Fig 2F). This indicates that the ability of OtUBD resin to enrich for ubiquitylated species is based on its binding affinity toward ubiquitin.

Taken together, these results indicate the OtUBD resin specifically enriches ubiquitin and ubiquitylated polypeptides as well as proteins that interact with ubiquitin-containing proteins.

## Purifications using OtUBD with denatured extracts enrich ubiquitin–protein conjugates

To distinguish proteins covalently modified by ubiquitin from proteins co-purifying through noncovalent interaction with ubiquitin or ubiquitylated proteins, we optimized pulldown conditions to include a denaturation step (Fig 3A). Yeast lysates were incubated with 8 M urea, a condition where the majority of proteins are unfolded, to dissociate protein complexes [48]. Denatured lysates were then diluted with native lysis buffer (to a final urea concentration of 4M) to facilitate the refolding of ubiquitin and applied to the OtUBD resin. A similar method was used previously in ubiquitin immunoprecipitation using the FK2 monoclonal antibody [20]. Under such conditions, the OtUBD resin concentrated ubiquitylated proteins with efficiencies similar to those seen under native conditions (Fig 3C). At the same time, the denaturing treatment greatly reduced the total amount of proteins eluted compared to native conditions, and the spectrum of purified protein species also changed (Fig 3D). This suggests that ubiquitylated proteins were specifically enriched by the urea treatment.

To verify that OtUBD pulldown following a denaturation step is specific for proteins covalently modified with ubiquitin, we utilized a yeast strain whose endogenous ubiquitin-coding sequences were all deleted and replaced with a single plasmid-borne $His_6$-tagged ubiquitin sequence [28]. The eluted fractions from OtUBD resin pulldowns performed after either denaturing or nondenaturing treatments of lysates (Fig 3A) were then denatured again by incubation with urea or guanidine-HCl (Fig 3B). The denatured proteins were applied to a $Co^{2+}$ (Talon) resin for immobilized metal affinity chromatography (IMAC) via the $His_6$-tagged ubiquitin. If the eluate from the OtUBD resin had contained only ($His_6$-)ubiquitylated proteins, most or all of the total proteins should bind to the resin. We observed that when OtUBD pulldowns were done following a denaturing lysate treatment, most of the eluted proteins were indeed bound to the $Co^{2+}$ resin (Fig 3E). By contrast, a large portion of proteins from a "native" OtUBD pulldown remained in the flow-through of the $Co^{2+}$ resin (Fig 3E). The overall levels of ubiquitylated species recovered, however, were comparable between the 2 treatments (Fig 3F). Consistent with these findings with bulk ubiquitin conjugates, when we tested whether the proteasome, which binds noncovalently to many polyubiquitylated substrates

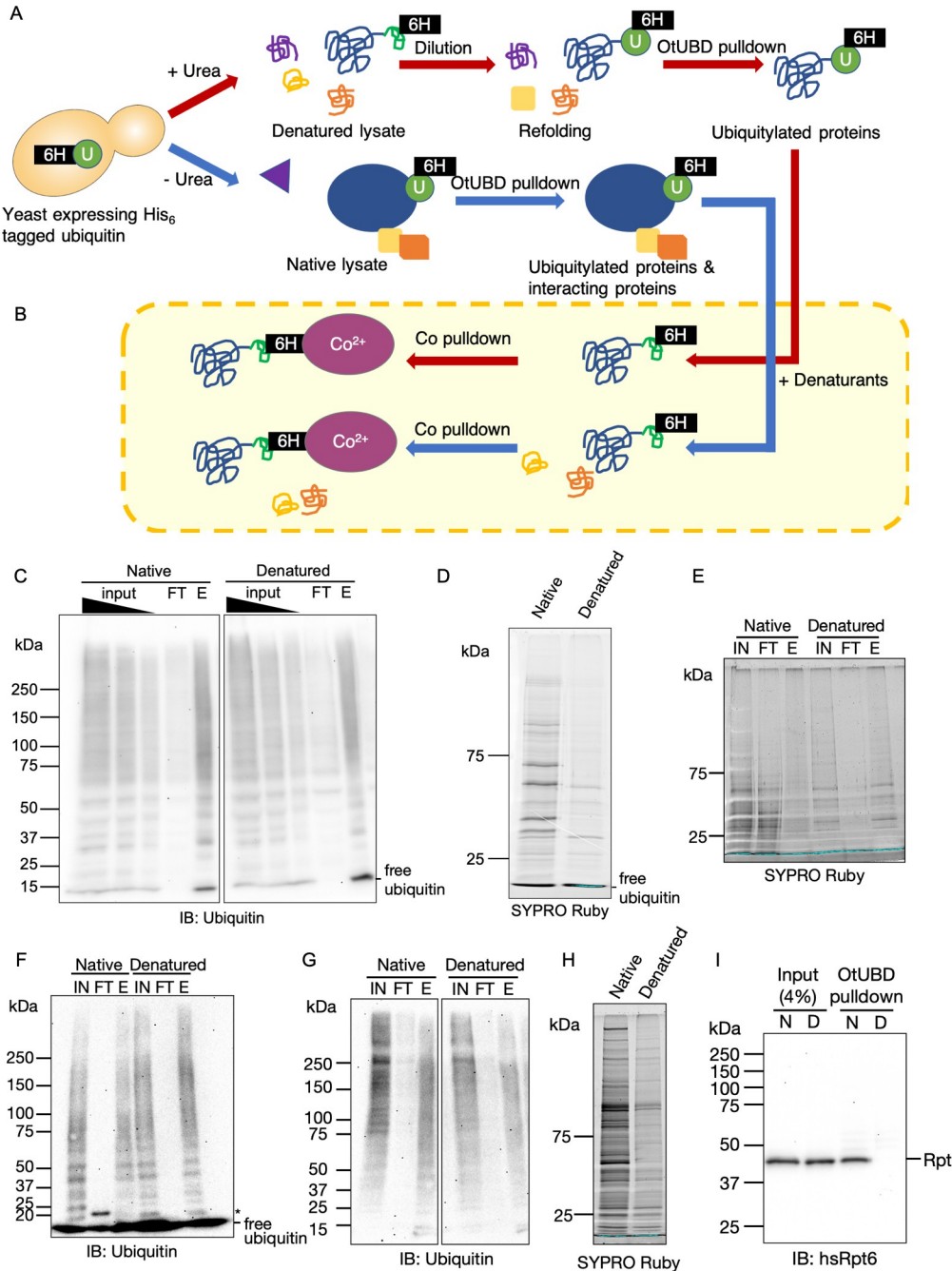

**Fig 3. OtUBD pulldown under denaturing condition specifically enriches for proteins covalently modified with ubiquitin.** **(A)** Workflow of OtUBD pulldowns following sample denaturation (red arrows) or under native (blue arrows) conditions. In the first case, cell lysate is treated with 8 M urea to denature and dissociate proteins. The denatured lysate is then diluted 1:1 with native buffer to allow ubiquitin to refold and bind to OtUBD resin. Under such conditions, only ubiquitylated proteins are expected to be enriched. In the second case, cell lysate contains native ubiquitylated proteins as well as proteins that interact with them. OtUBD pulldown under such conditions is expected to yield both ubiquitylated substrates and ubiquitin-binding proteins. **(B)** Outline for the use of tandem $Co^{2+}$ resin pulldowns to validate OtUBD pulldown results under different conditions. Eluates from OtUBD after lysates were incubated with denaturant (red arrows) or left untreated (blue arrows) are (re)treated with denaturant (8 M urea or 6 M guanidine•HCl) and then subjected to IMAC with a $Co^{2+}$ resin in denaturing conditions. Proteins covalently modified by $His_6$-ubiquitin bind to the $Co^{2+}$ resin while proteins that only interact noncovalently with ubiquitin end up in the flow-through. **(C)** Anti-ubiquitin blot of OtUBD pulldowns following native and urea denaturing treatments performed as described in Fig 3A. FT, flow-through; E, eluted fractions. The image was spliced to remove irrelevant

lanes. **(D)** Total protein present in eluates of the OtUBD pulldowns in Fig 3C visualized by SYPRO Ruby stain. **(E)** Total protein present in different fractions of the $Co^{2+}$ IMAC (see Fig 3B; the results shown here used urea as the denaturant) visualized by SYPRO Ruby stain. IN, input; FT, flow-through; E, fraction eluted with 500 mM imidazole. **(F)** Anti-ubiquitin blot of fractions from $Co^{2+}$ IMAC (see Fig 3B; the blot shown here used urea as the denaturant) of eluates from native and denaturing OtUBD resin pulldowns. IN, input; FT, flow-through; E, fraction eluted with 500 mM imidazole. *The identity of the prominent approximately 20 kDa species in the flow-through from the native extract is unknown. **(G)** Anti-ubiquitin blot of OtUBD pulldowns from HeLa cell lysates performed as described in Fig 3A following native or denaturing treatments. The image was spliced to remove irrelevant lanes. IN, input; FT, flow-through; E, fraction eluted with low pH elution buffer. **(H)** Total protein present in eluates in Fig 3G visualized by SYPRO Ruby stain. **(I)** Immunoblot analysis of human proteasomal subunit Rpt6 in OtUBD pulldowns following native and urea denaturing treatments of lysates. Unmodified Rpt6 co-purified with OtUBD resin under native conditions but not following denaturation of extract. N, native condition; D, denaturing condition.

[49], was in the OtUBD eluates, we readily detected unmodified proteasome subunits in the native pulldowns but not in pulldowns from denatured lysates (S2A Fig).

OtUBD-based affinity purifications, under either native or denaturing conditions, were also effective with human cell lysates. Both conditions led to similar enrichment of ubiquitin conjugates (Fig 3G), but the denaturing pretreatment greatly reduced the amounts of co-purifying nonubiquitylated proteins (Fig 3H). Congruent with this, nonubiquitylated human proteasomal subunits were only present at substantial levels in eluates from native lysates (Figs 3I and S2B). Interestingly, low amounts of presumptive ubiquitylated proteasome subunits were discovered in OtUBD pulldowns from both native and denatured lysates, and these species were strongly enriched over the unmodified subunits under the latter condition (S2B Fig).

Overall, these results indicate that OtUBD-based protein purification under denaturing conditions can specifically enrich proteins that are covalently modified by ubiquitin.

## Comparison of OtUBD with other ubiquitin-enriching reagents

Having developed the native and denaturing protocols for OtUBD-based ubiquitin enrichment, we decided to compare our new method side by side with existing ubiquitin enrichment reagents based on other UBDs or TUBEs. Dsk2 is a yeast protein that contains a ubiquitin-associated domain (UBA) at its carboxyl terminus [50]. Its ability to bind ubiquitylated proteins has been harnessed in the ubiquitin enrichment reagent GST-Dsk2, which has been successfully used to study the DNA damage-induced ubiquitylation of RNA polymerase-II [51,52]. To directly compare Dsk2 with OtUBD, we engineered an N-terminal cysteine residue as a handle to conjugate Dsk2 to SulfoLink resin. In addition, the aforementioned TUBE TR-TUBE [23], as well as the monomeric UBD it is based on (a trypsin resistant variant of the UBA domain of human Ubiquilin 1; we will call it TR-UBA) were similarly engineered for resin conjugation and comparison with OtUBD. (Coincidentally and to our advantage, none of these proteins originally contain a cysteine residue.)

After covalently attaching equimolar amounts of different ubiquitin affinity reagents to SulfoLink resin, we tested their respective abilities to enrich for ubiquitylated species from yeast whole cell lysates under both native and denaturing conditions. Under native conditions, Dsk2, TR-TUBE and OtUBD all efficiently pulled down the majority of ubiquitylated species from whole cell lysates when adequate amounts of resin were used; TR-UBA was much less efficient even when a much higher amount of resin was applied (Fig 4A). OtUBD and TR-TUBE resins displayed equal efficiency at the lowest resin amount tested. Although Dsk2 contains only a single UBD, it was able to efficiently purify poly-ubiquitylated proteins when sufficient amounts of resin were used, consistent with its high affinity toward poly-ubiquitin [50]. We also tested an old batch of OtUBD resin that had been stored at 4°C for about one year, and it performed similarly to the freshly prepared OtUBD resin, demonstrating an

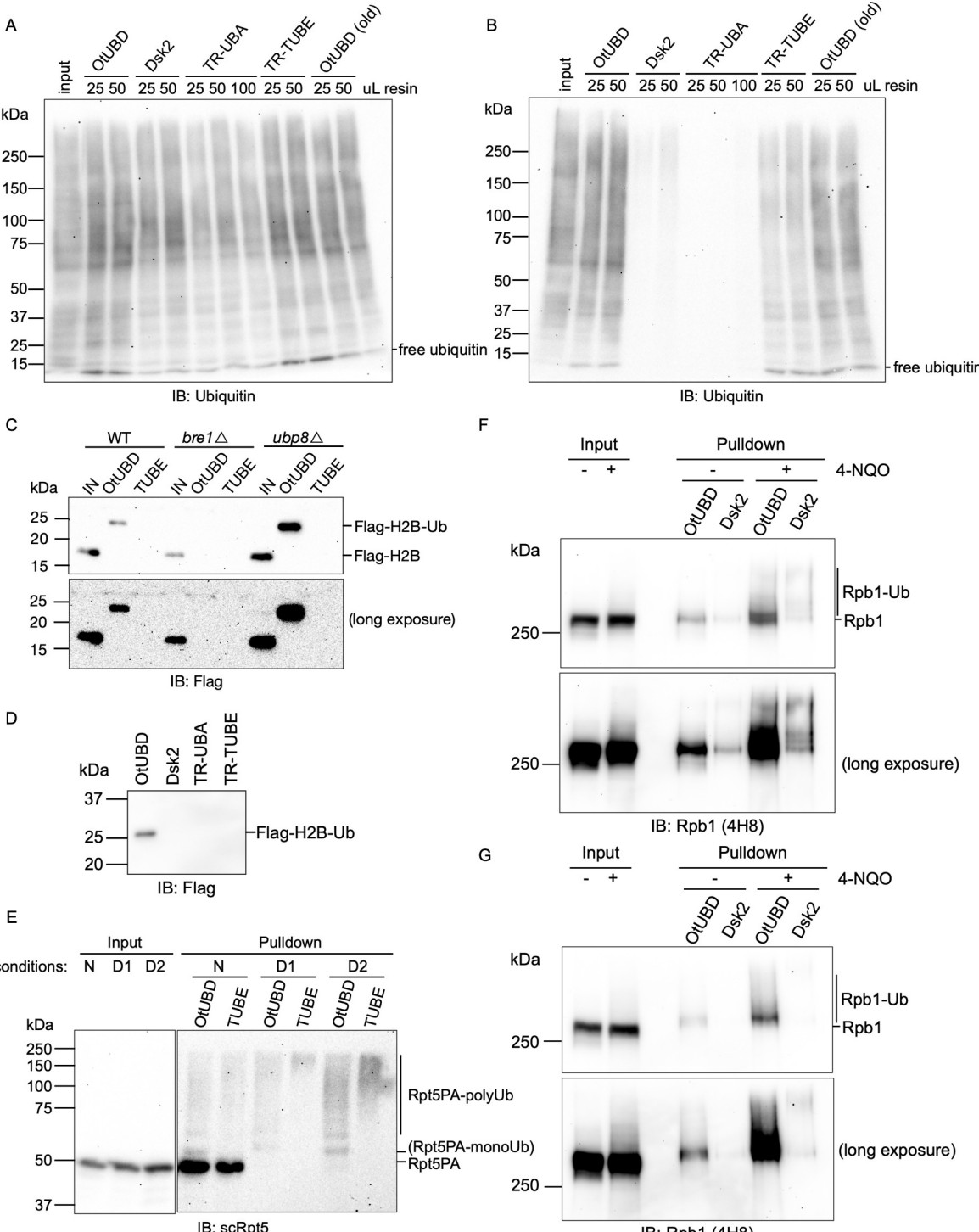

**Fig 4. Comparison of OtUBD with other ubiquitin enriching reagents. (A)** Anti-ubiquitin immunoblot of ubiquitin pulldowns from yeast whole cell lysate using different UBD/TUBE-based reagents under native conditions. Yeast whole cell lysate with 1mg of total protein was used in each pulldown. Elution was achieved by incubating resins with SDS sample buffer. **(B)** Anti-ubiquitin immunoblot of ubiquitin pulldowns from yeast whole cell lysate using different UBD/TUBE-based reagents following lysate denaturation. Yeast whole cell lysate with 1 mg of total protein was used in each pulldown. Elution was achieved by incubating resins with SDS sample buffer. **(C)** Anti-Flag immunoblot of OtUBD and TUBE pulldowns from lysates of WT, *bre1Δ* and *ubp8Δ* yeasts expressing Flag-tagged histone H2B. OtUBD resin but not TUBE bound monoubiquitylated histone H2B from whole cell lysate of WT and *ubp8Δ* yeasts. IN, input; UBD, eluted fraction from OtUBD pulldown; TUBE, eluted fraction from TUBE pulldown. Elution was achieved by incubating resins with SDS

sample buffer. **(D)** Anti-Flag immunoblot of pulldowns using different UBD/TUBE-based reagents from lysate of *ubp8Δ* yeast expressing Flag-tagged histone H2B. Only the OtUBD resin was able to bind detectably to monoubiquitylated histone H2B. **(E)** Western blot of yeast Rpt5 from pulldowns of *rpt2-P103A rpt5-P76A* (*rpt2,5PA*) mutant yeast lysates using different ubiquitin affinity resins under different conditions. N, pulldown performed under native conditions; D1, pulldown performed after 8 M urea treatment of lysate as described in Fig 3A; D2, pulldown performed with lysate directly extracted in buffer containing 8 M urea. **(F)** Western blot analysis of RNAPII subunit Rpb1 in OtUBD and Dsk2 pulldowns of native HeLa cell lysates. Cells were treated with 4-NQO to induce RNAPII ubiquitylation. **(G)** Western blot analysis of RNAPII subunit Rpb1 in OtUBD and Dsk2 pulldowns of denatured HeLa cell lysates. Cells were treated with 4-NQO to induce RNAPII ubiquitylation. TUBE, tandem ubiquitin-binding entity; UBD, ubiquitin-binding domain; WT, wild-type; 4-NQO, 4-nitroquinoline-1-oxide.

excellent shelf-life for this resin (Fig 4A, last 2 lanes). Strikingly, when the lysate denaturing pretreatments described earlier were used with the different resins, only OtUBD could efficiently enrich ubiquitylated species (Fig 4B). TR-TUBE only partially enriched ubiquitylated substrates compared to OtDUB under such conditions, while Dsk2 and TR-UBA barely pulled down any ubiquitylated proteins.

We then compared the different resins for their abilities to detect ubiquitylated forms of specific proteins of interest. Histone H2B is monoubiquitylated by the ubiquitin E3 ligase Bre1 and deubiquitylated by the DUB Ubp8 in yeast [45,53]. The monoubiquitylated derivative of histone H2B is difficult to detect directly in the whole cell lysate due to its low abundance in comparison to unmodified H2B (Fig 4C). To determine if the OtUBD resin could aid in the detection of monoubiquitylated H2B, we used OtUBD resin to purify total ubiquitylated proteins from cell lysates of wild-type (WT), *bre1Δ* and *ubp8Δ* yeast strains expressing Flag-tagged histone H2B and then analyzed the proteins by immunoblotting. A slower migrating band in the anti-Flag immunoblot, which represents the monoubiquitylated H2B, was detected in the WT and *ubp8Δ* yeasts but not in the *bre1Δ* yeast (Fig 4C). By contrast, the TR-TUBE-resin failed to capture the monoubiquitylated H2B in any of the yeast samples, including those with elevated levels of H2B monoubiquitylation due to deletion of *UBP8* (Fig 4C). We also tested whether any of the other resins we made were able to capture mono-ubiquitylated histone H2B in *ubp8Δ* yeast lysate and found that neither Dsk2 nor TR-UBA could (Fig 4D). Thus, although both Dsk2 and TR-TUBE-resin can efficiently enrich bulk ubiquitylated species from yeast lysates (Fig 4A), these reagents can be limited in their abilities to detect certain monoubiquitylated proteins. These results again highlight a potential advantage of OtUBD relative to other UBDs or TUBEs in studying monoubiquitylated substrates.

Our group previously identified Pro-to-Ala mutations in yeast proteasomal subunits Rpt2 and Rpt5 (strain "*rpt2,5PA*") that lead to their misfolding and ubiquitylation under normal growth conditions [54]. In that study, ubiquitylation of the Rpt subunits was confirmed by overexpressing His-tagged ubiquitin in the proteasome mutant yeast strain and performing IMAC under denaturing conditions to capture ubiquitylated species. We performed OtUBD and TUBE pulldowns with *rpt2,5PA* yeast lysates without ubiquitin overexpression. Based on anti-Rpt5 immunoblotting, both resins captured a smear of higher mass Rpt5PA species, which are likely endogenous polyubiquitylated Rpt5PA species (Fig 4E). Unmodified Rpt5PA co-purified with both OtUBD and TUBE under native conditions but was largely eliminated from both pulldowns following lysate denaturation. Under native conditions, both OtUBD and TUBE purified similar amount of ubiquitylated Rpt5PA (Fig 4E, lane 4 and 5) while under denaturing conditions, OtUBD captured more ubiquitylated species especially at the lower molecular weight range (Fig 4E, lane 6 to 9). Compared to the TUBE pulldown, OtUBD pulldown captured an additional slower-migrating Rpt5PA species which, based on the apparent molecular mass, is likely monoubiquitylated Rpt5PA. In this experiment, we also found that denatured lysate extracted directly with urea buffer (condition D2) gave more ubiquitylated Rpt5PA compared to when the lysate was extracted with a native buffer first and then

denatured with added urea (condition D1). The former method is expected to solubilize ubiquitylated species that would normally precipitate.

As a final example of single protein analysis, we used OtUBD to detect ubiquitylated RNA polymerase II (RNAPII) in cultured human cells. RNAPII becomes ubiquitylated upon UV-induced DNA damage [55]. Rpb1, the largest subunit of RNAPII, is heavily ubiquitylated under such conditions [56]. We treated HeLa cells with the chemical 4-nitroquinoline-1-oxide (4-NQO), which mimics the biological effects of UV on DNA (57), and performed OtUBD and Dsk2 pulldowns of both native (Fig 4F) and denatured (Fig 4G) lysates with saturating amounts of each resin (as determined in Fig 4A). In both cases, OtUBD pulldown captured similar slower migrating bands when analyzed by anti-Rpb1 immunoblotting. Because Rpb1 is a large protein of over 200 kDa and exists in different phosphorylation states, it is difficult to distinguish nonubiquitylated and monoubiquitylated species based on migration through an SDS-PAGE gel [58]. OtUBD pulldown under denaturing conditions, in this case, provides confidence that Rpb1 is ubiquitylated even under basal conditions and becomes heavily ubiquitylated upon treatment with 4-NQO (Fig 4G). Dsk2 pulldown captured much less ubiquitylated Rpb1 under both conditions compared to OtUBD. The 2 reagents may potentially capture different populations of ubiquitylated Rpb1 based on the appearance of the bands observed (Fig 4F).

These examples illustrate how OtUBD resin can facilitate the detection of monoubiquitylated and polyubiquitylated proteins in both yeast and human cells. Importantly, in several different examples, OtUBD demonstrated its advantages over other UBD/TUBE-based reagents examined, particularly in its ability to capture monoubiquitylated proteins and to bind ubiquitylated species following protein denaturation.

## OtUBD-pulldown proteomic profiling of the yeast and human ubiquitylome and ubiquitin interactome

By comparing OtUBD pulldowns of native and denatured cell lysates, we can potentially differentiate different ubiquitin-related proteomes in a biological sample. The "ubiquitylome," i.e., the collection of covalently ubiquitylated proteins, can be defined as the protein population eluted from an OtUBD affinity resin used with denatured cell extracts. The "ubiquitin interactome" can be roughly defined as those proteins that are specifically enriched following OtUBD pulldowns from native extracts but not pulldowns from denatured lysates (Fig 3A). Notably, the latter definition will exclude cases where a subpopulation of a protein is ubiquitylated while the nonubiquitylated population of the same protein interacts noncovalently with ubiquitin or other ubiquitylated proteins. For example, some proteasomal subunits are known to be ubiquitylated [59], but proteasome particles where these subunits are unmodified still interact noncovalently with ubiquitylated proteins. Proteins such as these proteasomal subunits will be excluded from the ubiquitin interactome as defined here. Nevertheless, these definitions provide a general picture of the ubiquitylome and ubiquitin interactome.

We performed OtUBD pulldowns of whole yeast lysates with and without prior denaturation (S3A–S3C Fig) and profiled the eluates using shotgun proteomics. The amount of resins used was predetermined empirically to avoid saturation of binding capacity. For each condition, we included 2 biological replicates and for each biological replicate, 2 technical repeats of the LC-MS/MS run. Control pulldowns by SulfoLink resin without OtUBD were performed in parallel to eliminate proteins that nonspecifically bind to the resin. As was seen in earlier experiments, the control pulldowns yielded no detectable ubiquitylated species and only trace amounts of proteins (S3A–S3C Fig). Some proteins were identified in a subset of the control pulldown replicates (S3D Fig), due partially to carryover of high abundance peptides from

previous runs, but the overall quantities of proteins in these control samples, as demonstrated by TIC (total ion current), were much lower compared to the OtUBD pulldown samples (S3E Fig). Hence, for each biological replicate, only proteins present at significantly higher levels (>20-fold) in the OtUBD pulldown samples over the corresponding control pulldown samples were considered real hits (S3F Fig and S2 Data).

The 2 pulldown conditions yielded similar total numbers of proteins (Fig 5A) with a major overlap of protein identities. Over 400 proteins were discovered exclusively under native conditions, suggesting they are not ubiquitylated themselves but co-purify with ubiquitin or ubiquitylated proteins. Interestingly, over 600 proteins were identified only under denaturing conditions. Because OtUBD pulldowns following lysate denaturation yield much less total protein than under native conditions (S3C Fig), the possibility of identifying low abundance proteins in the LC-MS/MS analysis is likely increased.

We compared the OtUBD-defined yeast ubiquitylome with data from previously published studies using di-Gly remnant antibody-based methods (Fig 5B) [60–62]. Our study identified 1,811 ubiquitylated yeast proteins, the second highest number among the 4 studies compared here. About two-thirds of proteins identified in our study have been reported to be ubiquitylated by at least one of these di-Gly antibody-based studies. Around 600 ubiquitylated proteins were uniquely identified in this study. Some of these might involve noncanonical ubiquitylation, where the ubiquitin modifier is covalently attached to a nucleophilic residue on the substrate other than a lysine [3].

GO analysis indicated that the yeast ubiquitylome defined by OtDUB binding spans proteins from a wide variety of cellular processes, including multiple biosynthesis pathways, protein localization, vesicle-mediated transport, and protein quality control pathways (Fig 5C). By contrast, the ubiquitin interactome, as defined above, appeared to yield greater representation in nucleic acid-related processes such DNA replication, RNA transcription, ribosome biogenesis, and noncoding RNA processing (Fig 5D).

We also performed OtUBD pulldowns with denatured HeLa cell lysates alongside immunoprecipitations using the FK2 anti-ubiquitin monoclonal antibody, which has been used successfully with a similar denaturation protocol [20]. Three biological replicates, each with 2 technical repeats of LC-MS/MS runs, were analyzed. OtUBD resins efficiently and consistently enriched ubiquitylated proteins from HeLa cell lysates. By contrast, we observed that different batches of FK2 antibodies gave different efficiencies of enrichment; this is evident in the amount of ubiquitylated species present in the flow-through fractions of FK2 immunoprecipitations in 2 of the biological replicates using different batches of antibodies (S4A and S4B Fig) as well as the total number of proteins identified in each biological replicate. This could be a result of inconsistency in antibody quality across batches or in efficiency of antibody immobilization. The majority of ubiquitylated proteins identified by the FK2 antibody resin were also found in the ubiquitylated proteome identified by OtUBD resin (Fig 5E). Compared to the FK2 immunoprecipitations, OtUBD pulldowns identified about 1,000 additional ubiquitylated proteins, the majority of which have at least one reported ubiquitylation site in a previous study using diGly antibodies [63].

We also compared our results with published data based on TUBE or diGly antibody-based enrichment methods. For human ubiquitylome discovery, different studies using the same enrichment method yielded a wide range in the total number of detected proteins [35]. As the human ubiquitylome is much more complex than the yeast ubiquitylome, proteomics methods and practices that can affect the depth of protein identifications will more strongly influence the overall results with human cells (see example in Discussion). Therefore, comparing the number of identified proteins across different studies will reflect not only the relative

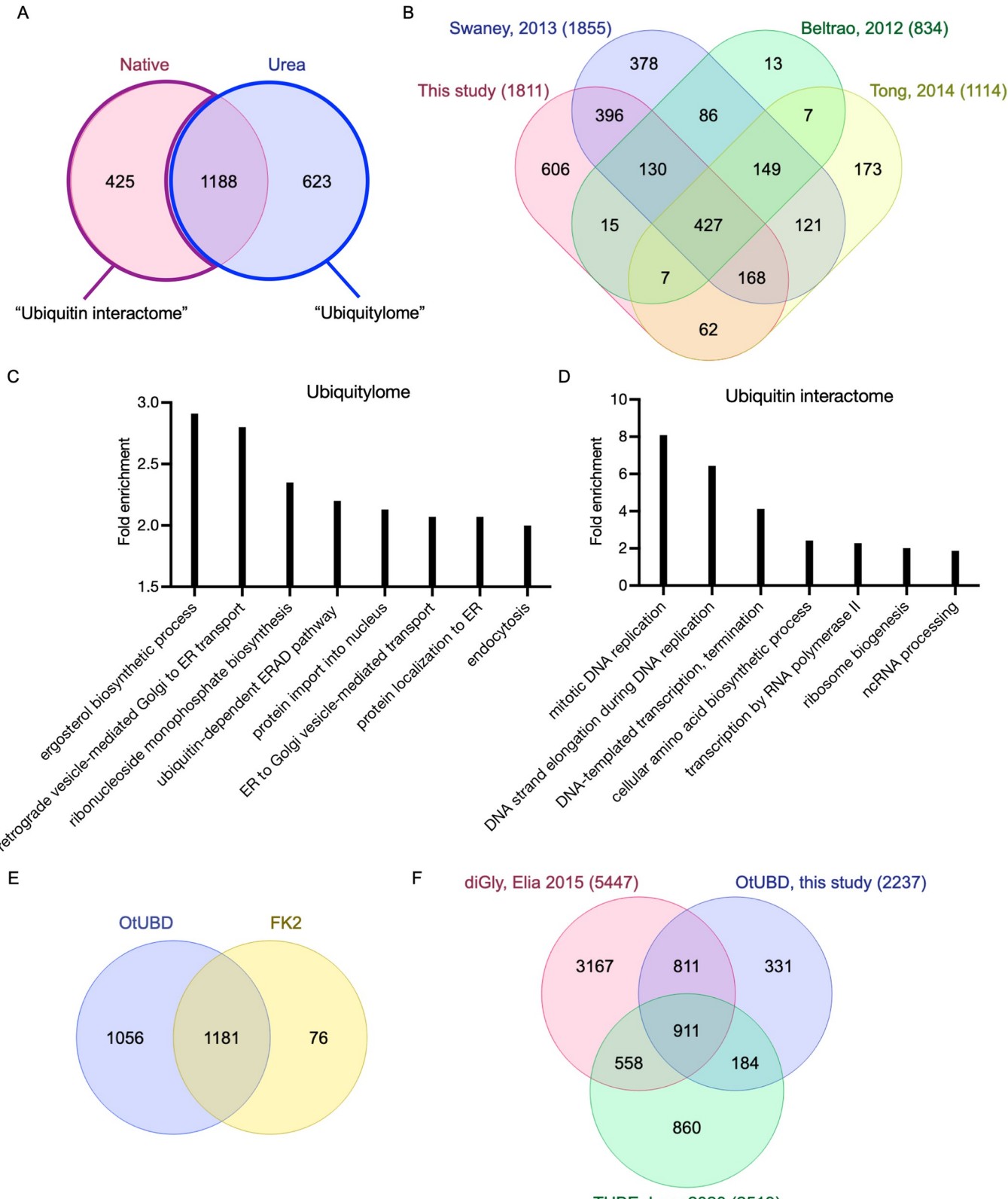

**Fig 5. OtUBD pulldown-proteomics enables profiling of the ubiquitylome and ubiquitin interactome of yeast and human cells. (A)** Venn diagram of yeast proteins identified by OtUBD pulldown-proteomics with the pulldowns performed following either nondenaturing or urea denaturing treatments. The

collection of proteins identified in OtUBD pulldowns under denaturing conditions is defined as the ubiquitylome (blue outline). The collection of proteins identified only in native OtUBD pulldowns is defined as the ubiquitin interactome (purple outline). **(B)** Venn diagram comparing the yeast ubiquitylome defined by OtUBD pulldown-proteomics with 3 previous studies using the di-Gly antibody IP method [60–62]. The numbers listed in the brackets are the total number of ubiquitylated proteins identified in each study. **(C, D)** Top biological pathways involved in OtUBD pulldown-defined yeast ubiquitylome (C) and ubiquitin interactome (D) based on GO analysis. The numeric values supporting these panels can be found in S3 Data. **(E)** Venn diagram comparing the human ubiquitylome defined by OtUBD pulldown and FK2 antibody IP performed in the current study. **(F)** Venn diagram comparing the published human ubiquitylomes defined by diGly antibody [63] and TUBE-based enrichments [64] and the OtUBD-defined ubiquitylome obtained in this study. The numbers listed in the brackets are the total number of ubiquitylated proteins identified in each study. GO, Gene Ontology; TUBE, tandem ubiquitin-binding entity.

efficiencies of the different ubiquitin enrichment methods but also differences in the proteomics methodology and cell culture conditions.

With this caveat in mind, we compared our results in terms of total number of identifications and the identities of ubiquitylated proteins with published data sets describing the human ubiquitylome using the diGly antibody enrichment [63] and TUBE methods [64], respectively (Fig 5F). The diGly-based result from Elia and colleagues, which discovered the largest number of ubiquitylated proteins among different diGly literatures we examined, encompassed a much higher total number of ubiquitylated proteins compared to our study and the TUBE-based study, which yielded similar numbers of total proteins. Compared to the TUBE-based identifications, our data overall showed greater overlap with the diGly antibody-based result—77% of proteins detected in our analysis and 58% of those in the TUBE study were supported by at least one diGly-remnant-containing peptide reported in the diGly antibody-based study. One explanation for the lower congruence for the TUBE-based enrichment was that it was done under native conditions and therefore likely included many nonubiquitylated polypeptides that co-purified with ubiquitin or ubiquitylated proteins.

These experiments demonstrate that the OtUBD affinity resin can be used to profile the ubiquitylated proteome of both yeast and human cells. Moreover, all 7 lysine ubiquitin–ubiquitin linkages were identified in the yeast proteomics (S4C Fig), and their ratios roughly agreed with previous quantitative studies of relative linkage frequencies [4], indicating relatively unbiased enrichment of these different lysine linkages. The molecular basis of this apparent indifference to chain linkage is uncertain. Previous analysis by isothermal calorimetry of OtUBD binding to monomeric ubiquitin revealed a 1:1 complex with a $K_d$ of approximately 5 nM; ubiquitin dimers linked by either K48 or K63 bound extremely tightly, preventing accurate determination of their dissociation constants by this method [41]. OtUBD might bind to the distal ubiquitin in all chain types with high affinity, leading to unbiased enrichment of different polyubiquitin chains, as well as detection of noncanonical ubiquitylation. All except the K33 ubiquitin–ubiquitin linkage were also identified in the HeLa cell proteomics. We note that K33 is a low abundance linkage [4], and employment of techniques that can increase the depth of identifications, such as an orthogonal fractionation step prior to LC-MS/MS, would likely allow its detection.

## OtUBD and label-free quantitation enable identification of potential E3 ligase substrates

Finally, we sought to apply OtUBD-pulldown proteomics for identifying substrates of specific E3 ubiquitin ligases. Identification of substrates for particular E3 ligases can be challenging due to the transient nature of E3-substrate interactions and the low abundance and instability of many ubiquitylated proteins [65]. One way to screen for potential substrates is to compare the ubiquitylome of cells with and without (or with reduced level/function of) the E3 of interest [66]. Proteins with higher ubiquitylation levels in the cells expressing the E3 compared to cells lacking it would be candidate substrates. We used OtUBD-pulldown proteomics to

compare the ubiquitylomes of wildtype BY4741 yeast and 2 congenic E3 deletion strains, *bre1Δ* and *pib1Δ*, obtained from a yeast gene knockout library [67]. Bre1 is a relatively well-characterized E3 ligase that monoubiquitylates histone H2B [53]. This ubiquitylation does not lead to H2B proteolysis but is involved in important chromosomal processes, including transcription and DNA damage repair [68]. Other substrates of Bre1 are largely unknown [69]. Pib1 is a much less studied E3 ligase that localizes to endosomes and the vacuole and participates in endosomal sorting [70].

We harvested WT, *bre1Δ*, and *pib1Δ* yeast cells and following lysate denaturation, performed OtUBD pulldowns. Proteins eluted from the OtUBD resin were subject to label-free quantitative proteomics (Figs 6A, S5A and S5B). Three biological replicates were examined for each group, and each replicate was analyzed by 2 separate LC-MS/MS runs. Quantitation was performed using total TIC after normalization among the analyzed samples. As expected, histone H2B (identified as Htb2) presented at a much higher level in the ubiquitylome of WT cells compared to that of *bre1Δ* cells (Fig 6B). Interestingly, we identified 2 different ubiquitylation sites on histone H2B (Htb2) in different samples (Figs 6C, S6A and S6B). The K123 ubiquitylation site, which is the major reported ubiquitylation site of Bre1 on histone H2B [71,72], was detected in WT cells but not in *bre1Δ* cells. By contrast, the other ubiquitylation site, K111, showed up in both WT and *bre1Δ* cells. This suggests there is an E3 ligase(s) other than Bre1 that ubiquitylates histone H2B on K111. Although this ubiquitylation site had been reported in a diGly antibody-based proteomics study [61], its function remains to be studied. Besides histone H2B, we also identified 16 other proteins present in significantly higher levels in the WT cell ubiquitylome compared to *bre1Δ* cells (Fig 6D). In addition, 35 proteins were exclusively detected in the ubiquitylome of WT cells (Fig 6E). Taken together, these proteins are considered potential Bre1 substrates. Interestingly, some of these proteins (Fig 6D and 6E, green) had been shown to be metabolically stabilized in *bre1Δ* cells in an earlier study [73], which indicated that they could be direct or indirect proteolytic ubiquitylation substrates of Bre1.

Analogous to the Bre1 data, we identified 3 proteins whose ubiquitylated forms were found at significantly higher levels in WT cells versus *pib1Δ* cells (Fig 6F and 6G) and 38 proteins that were detectably ubiquitylated in WT cells but not *pib1Δ* cells (Fig 6H). Of these proteins, 6 have been shown previously to be stabilized in *pib1Δ* cells (Fig 6G and 6H, green) [73].

Whether these potential E3 substrates are direct or indirect ubiquitylation substrates of the tested E3s will need to be validated by biochemical assays. Nonetheless, our results demonstrated that OtUBD can be used to profile ubiquitylomes quantitatively, which will be useful in the identification of novel substrates for E3 ligases and other ubiquitin-related enzymes such as E2s and DUBs.

Among the various proteomics data obtained from our OtUBD pulldowns, we observed a number of potential nonlysine ubiquitylation sites assigned by the Mascot search algorithm (S2 Data), substantiating the idea that OtUBD likely can enrich proteins with nonlysine ubiquitylation sites. (Please note that in this list, the GG-peptide assignments by the search algorithm have not been validated and will include some inaccurate assignments. Further validation is required to confirm any individual assignment.) We confirmed one of these sites by manual validation of the spectrum assignment (S6C Fig).

## Discussion

Protein ubiquitylation continues to be of great interest due its vital contributions to many fundamental cellular processes and for its important roles in human disease. Many enzymes involved in ubiquitylation are being pursued as targets for therapeutics [74,75]. For example, a number of drug candidates targeting E3 ligases such as MDM2 and XIAP have entered clinical trials for treatment of multiple types of cancer [76]. A variety of reagents and methods to study

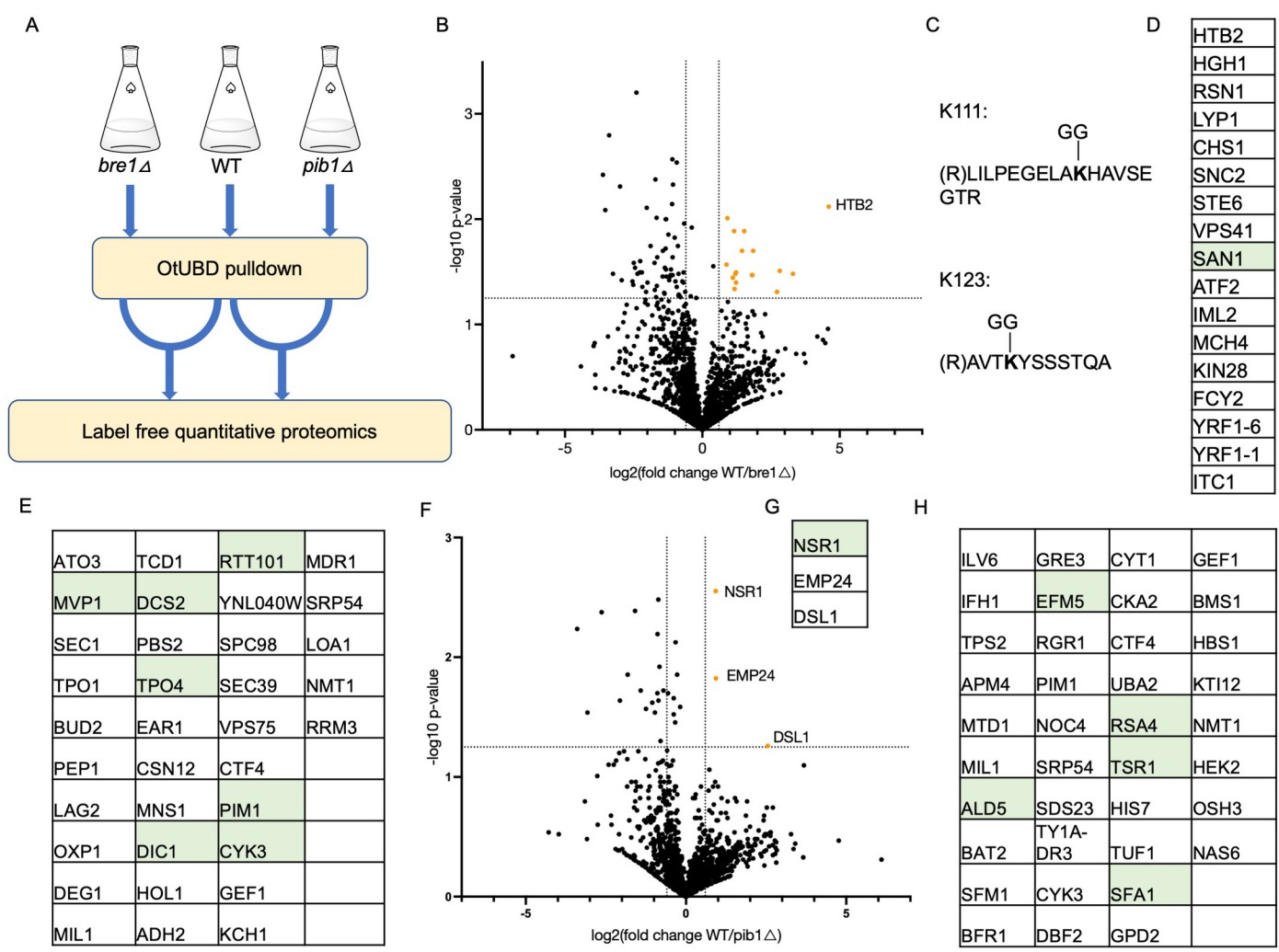

**Fig 6. Identification of potential E3 substrates by OtUBD pulldown and label-free quantitation. (A)** Scheme for E3 substrate identification using OtUBD pulldown and quantitative proteomics. WT and E3 deletion (*bre1Δ* and *pib1Δ*) yeast strains were subjected to OtUBD pulldowns following extract denaturation. The eluted proteins were then analyzed by label-free quantitation. **(B)** Volcano plot comparing WT and *bre1Δ* samples. Orange dots represent proteins that were significantly enriched in WT samples compared to *bre1Δ* samples. Horizontal dashed line indicates *p* = 0.05. Vertical dash lines indicate relative change of +/− 1.5-fold. The numeric values supporting this figure can be found in S2 Data. **(C)** Two different ubiquitylation sites identified on histone H2B (Htb2) in different samples. **(D)** List of proteins that were significantly enriched in WT samples compared to *bre1Δ* samples (orange dots in B). Green color indicates proteins that were previously reported to be stabilized in *bre1Δ* yeast. **(E)** Ubiquitylated proteins detected exclusively in WT but not *bre1Δ* samples. Green color indicates proteins previously reported to be stabilized in *bre1Δ* cells. **(F)** Volcano plot comparing WT and *pib1Δ* samples. Orange dots represent proteins that were significantly enriched in WT samples compared to *pib1Δ* samples. Horizontal dash line indicates *p* = 0.05. Vertical dash lines indicate relative change of +/− 1.5-fold. The numeric values supporting this figure can be found in S2 Data. **(G)** List of proteins that were significantly enriched in WT samples compared to *pib1Δ* samples (orange dots in F). Green color indicates proteins that were previously reported to be stabilized in *pib1Δ* yeast. **(H)** Ubiquitylated proteins detected exclusively in WT but not *pib1Δ* samples. Green color indicates proteins previously reported to be stabilized in *pib1Δ* yeast. WT, wild-type.

ubiquitylation or ubiquitylation-related processes have been developed, but these methods all have limitations [16,39]. For example, TUBEs are effective at detecting polyubiquitin chains, but this creates a bias toward polyubiquitylated substrates; they often fail to detect protein monoubiquitylation signals (e.g., Fig 4C and 4D), which can dominate the ubiquitylome in at least some mammalian cell types [17]. As a result, new and economical reagents and methods to analyze the many types of ubiquitin modification are still needed, particularly when these modifications are present at very low levels.

The versatile high-affinity UBD domain of OtDUB provides a ubiquitin affinity reagent with several advantages over existing tools. First, it is straightforward and relatively inexpensive to generate the affinity resin using the small recombinant OtUBD protein expressed and purified from *Escherichia coli*. Second, ubiquitin enrichment using OtUBD is applicable to both monoubiquitylation and polyubiquitylation, in contrast to the bias of TUBEs and other UBD-based reagents such as the ones we tested (Fig 4D). Third, OtUBD pulldowns can be performed under native conditions for the study of both ubiquitylated substrates and proteins that associate noncovalently with them; alternatively, by subjecting extracts to denaturing conditions prior to pulldown, OtUBD pulldowns can be tuned toward proteins covalently modified by ubiquitin. We demonstrated that OtUBD pulldowns, coupled with proteomics, can be used to profile the ubiquitylated proteomes of yeast and mammalian cells; it can undoubtedly be used to characterize the ubiquitylomes of other eukaryotic cells as well. Fourth, comparative OtUBD pulldown-proteomics can be used to identify substrates of ubiquitylating enzymes (E2s or E3s), as shown here, or DUBs. Finally, unlike the anti-diGly immunoaffinity tool that is specific for diGly remnants on Lys side chains, OtUBD-based purifications can potentially help identify noncanonical ubiquitin–protein linkages such as those through Cys, Ser, or Thr side chains, the N-terminal amino group, or chemical bonds that do not involve the ubiquitin carboxyl terminus, as in ubiquitylation mediated by *Legionella* SidE proteins [3,77,78]. It should also be possible to enrich ubiquitin linkages to macromolecules other than proteins, such as the recently discovered ubiquitin-lipopolysaccharide adducts formed during *Salmonella* infections [40].

We have demonstrated that OtUBD is specific toward ubiquitin and ubiquitylated proteins. However, several caveats should be noted. Although OtUBD pulldowns following extract denaturation significantly reduces the amount of interacting proteins co-purifying with ubiquitylated proteins, a small number of noncovalently interacting proteins may still be co-purified in some cases (e.g., S2B Fig). Additional stringent wash steps may help mitigate this problem. OtUBD also binds to the closely related ubiquitin-like modifier Nedd8, although with a much lower affinity than for ubiquitin [41]. Like ubiquitin, Nedd8 is used for protein posttranslational modification [79] and because they leave the same -GG remnant after trypsin digestion, it is hard to differentiate the 2 modifiers using the normal diGly antibody method [16]. We looked for potential neddylation substrate(s) in our proteomics studies. Rub1 (yeast Nedd8), Cdc53, Rtt101, and Cul3 (3 yeast cullin proteins reported to undergo neddylation [80]) were detected in the OtUBD-defined ubiquitylome, which may have been enriched on the OtUBD resin through Nedd8 binding. Nevertheless, neddylation occurs at much lower levels compared to ubiquitylation [37] and based on the specificity analysis we performed (Figs 2D and 3E), neddylated proteins (if any) should account for only a small fraction of the OtUBD-enriched proteome.

In our OtUBD pulldown-proteomics experiments, the total number of ubiquitylated yeast proteins identified was comparable to previous studies using the di-Gly antibody enrichment method [60–62]. For the analysis of human cells, OtUBD pulldowns performed better than FK2 antibody immunoprecipitations in our hands, both in terms of overall efficiency and repeatability. Inconsistency in the performance of antibodies from different batches has been a widely recognized issue [81]. Recombinant OtUBD, by contrast, appeared to be consistent across different batches as well as stable during long-term storage. As the human ubiquitylome is much more complex compared to the yeast ubiquitylome, proteomics identification depth greatly influences the total number of ubiquitylated proteins identified in each study [35]. Therefore, it is difficult to interpret results comparing our human ubiquitylome data with published data from other studies.

Optimization of our proteomics pipeline would likely increase the number of identified ubiquitylated proteins and ubiquitylation sites, especially for low-abundance proteins. In a paper benchmarking a TUBE-based enrichment method called ThUBD with human cell lysates [46], the authors initially fractionated the eluted proteins by SDS-PAGE prior to trypsin digestion and LC-MS/MS; they identified 1,663 ubiquitylated proteins in their TUBE enriched samples. Switching to an orthogonal LC-based fractionation of digested peptides prior to the LC-MS/MS run, the authors were able to increase their overall identification to over 7,000 proteins, highlighting the strong influence of proteomics methodologies in ubiquitylome characterization. In the samples we analyzed, peptides derived from ubiquitin accounted for a significant percentage of the total number of identified peptides. This likely suppressed detection of low-abundance peptides, especially those with similar retention times. Preclearing of free ubiquitin from the eluted samples before LC-MS/MS, for example, by gel separation or an affinity depletion specific for free ubiquitin [82], would likely reduce this problem. Alternatively, fractionation of the protein or peptide samples prior to LC-MS/MS should also enhance the overall discovery rate.

OtUBD-based ubiquitin purification could be used in conjunction with methods that efficiently enrich certain ubiquitylated species. For example, OtUBD pulldowns could be performed before di-Gly antibody immunoprecipitation. When performed against the enormous pool of peptides derived from the entire cell proteome, di-Gly antibody IP often needs to be done in multiple batches or for multiple rounds to ensure efficient enrichment [63]. A preliminary OtUBD pulldown step would significantly enrich for ubiquitylated substrates in the sample without creating any bias toward polyubiquitylated species. This will greatly increase the percentage of di-Gly-linked peptides present in the digested sample. Since OtUBD has exceptionally high affinity toward free ubiquitin, it could also be used with the Ub-Clipping technique [83]. In Ub-Clipping, ubiquitylated proteins are cleaved at the ubiquitylation sites by the protease Lb-Pro to generate diGly-linked monoubiquitin species and free ubiquitin$_{1-74}$. These species carry information on ubiquitin chain topology and posttranslational modifications of ubiquitin that can be deciphered by MS analysis. Deployment of OtUBD for other applications can be readily envisioned.

With the characterization of OtUBD-ubiquitin binding and crystal structures of OtUBD available [41], one can imagine further modifications that would adapt or enhance OtUBD for other uses. For example, directed evolution or structure-based rational mutagenesis may be performed to change OtUBD binding specificity toward ubiquitin, specific ubiquitin chains or ubiquitin-like proteins. OtUBD could be used to make other ubiquitin detection tools by attaching a fluorophore or other functional handles to it. As a recombinant protein reagent that is versatile and easy to prepare, OtUBD will be an economical addition to the ubiquitin research toolbox.

## Materials and methods

### Plasmids and DNA cloning

The coding sequence for 3xOtUBD was synthesized by Genscript USA. pRSET-4xTR-TUBE was a gift from Yasushi Saeki (Addgene plasmid # 110312) [43]. The pRT498 vector, a bacterial expression plasmid modified from pET42b to include an N-terminal His$_6$-MBP with a cleavable TEV site, was used for expression of MBP and MBP-fusion proteins made in our lab. pET21a and pET42b vectors were used to express His$_6$-tagged proteins in bacteria. Plasmids and primers used in this study as well as insert sequences are described in detail in S1 Data. All PCR reactions were done using Phusion High-Fidelity DNA Polymerase (New England Biolabs, MA, USA).

## Yeast strains and growth

Yeast strains used are listed in S1 Data. Yeast cultures were grown overnight in yeast extract-peptone-dextrose (YPD) medium to saturation. The next day, the culture was diluted in fresh YPD to an $OD_{600}$ of 0.1 to 0.2 and cultured at 30°C with shaking until reaching mid-exponential phase ($OD_{600}$ 0.8 to 1.2). Cells were pelleted, washed with water, and flash frozen in liquid nitrogen and stored at −80°C until used.

## Mammalian cell culture

HeLa and HEK293T cells (ATCC) were cultured in Dulbecco's Modified Eagle Medium (Gibco) supplemented with 10% fetal bovine serum (Gibco) and 1% penicillin-streptomycin (Gibco). Cells were not used past 20 passages. To harvest cells for experiments, the medium was removed, and cells were washed with cold Dulbecco's phosphate buffered saline (DPBS, Gibco) before dislodging by scraping in cold DPBS. The dislodged cells were pelleted by centrifugation at $400 \times g$ for 4 minutes and flash frozen in liquid nitrogen. Cell pellets were stored at −80°C until used.

## Expression and purification of recombinant proteins

Recombinant His6-tagged proteins were purified from Rosetta (DE3) competent *E. coli* cells (Novagen, Germany) transformed with the appropriate plasmids. Bacterial cells were grown overnight in Luria-Bertani (LB) broth supplemented with either 100 μg/mL ampicillin (for pET21a-based plasmids) or 50 μg/mL kanamycin (for pRT498- and pET42b-based plasmids) and diluted 1/100 the next morning in fresh LB broth supplemented with the corresponding antibiotics. When cell density had reached 0.5 to 0.8 $OD_{600}$, protein production was induced by addition of isopropyl β-D-1-thiogalactopyranoside (IPTG) to a final concentration of 0.3 mM, and cells were cultured at 16 to 18°C overnight with shaking. Bacteria were pelleted and resuspended in bacteria lysis buffer (50 mM Tris•HCl, pH 8.0, 300 mM NaCl, 10 mM imidazole, 2 mM freshly added phenylmethylsulfonyl fluoride (PMSF)) supplemented with lysozyme and DNaseI, incubated on ice for 30 minutes and lysed using a French press. For purification of His6-tagged proteins and His6-MBP-tagged proteins, the lysates were clarified by centrifugation for 1 hour at 4°C at 10,000 rcf before being subjected to Ni-NTA (QIAGEN, Germany) affinity purification following the manufacturer's protocol. For further purification of His6-tagged OtUBD, Dsk2, TR-UBA, or 4xTR-TUBE, the protein eluted from the Ni-NTA matrix was supplemented with 5 mM tris(2-carboxyethyl)phosphine (TCEP) (from a 1M TCEP stock neutralized with NaOH to pH 7), concentrated, and then fractionated by Fast Protein Liquid Chromatography (FPLC) on a Superdex 75 gel filtration column (Cytiva, MA, USA) using FPLC buffer (50 mM Tris•HCl, pH 7.5, 150 mM NaCl, 1 mM TCEP). For further purification of His6-MBP-tagged proteins, the protein eluate from the Ni-NTA resin was concentrated and fractionated by FPLC on a Superdex 75 column using 50 mM Tris•HCl, pH 7.5, 150 mM NaCl buffer supplemented with 1 mM dithiothreitol (DTT).

For purification of OtUBD variants using pRT498-based plasmids, the His6-MBP-tagged proteins eluted from the Ni-NTA resin were subject to buffer exchange in a 50 mM Tris•HCl, pH 7.5, 150 mM NaCl buffer supplemented with 10 mM TCEP using a centrifugal filter device (Amicon, Germany, 3000 MWCO) following manufacturer's protocol. His-tagged TEV protease was added to remove the His6-MBP tag, and the mixture was incubated on ice overnight. The cleavage mixture was then allowed to flow through a column of clean Ni-NTA resin to capture the cleaved His6-MBP tag. The flow-through was concentrated and purified by FPLC with a Superdex 75 column using FPLC buffer.

The M48 DUB protein was prepared as described earlier [47].

All proteins were flash-frozen in liquid nitrogen and stored at −80˚C until use. Protein concentrations were determined by either SDS-PAGE and GelCode Blue (Thermo, USA) staining or a BCA assay (Thermo) using bovine serum albumin (BSA) as the standard.

## Immunoblotting and antibodies

Proteins resolved through SDS-PAGE gels were transferred to Immobilon-P PVDF membranes (Millipore, MA, USA) and blocked with 3% nonfat milk in Tris-buffered saline (20 mM Tris, 150 mM NaCl, pH adjusted to 7.5 with HCl) with 0.1% Tween-20 (TBST). Membranes were incubated first with the desired primary antibody diluted in TBST containing 1% nonfat milk for 1 hour at room temperature or overnight at 4˚C, washed extensively, and then incubated with an HRP-linked secondary antibody diluted in TBST with 1% milk for 1 hour at room temperature or overnight at 4˚C.

Primary antibodies used in this study were rabbit polyclonal anti-ubiquitin antibody (Dako, Denmark discontinued, 1:2,000 dilution), monoclonal mouse anti-ubiquitin antibody P4D1 (Enzo, USA, 1:1,000), monoclonal mouse anti-Flag M2 (Sigma, USA, 1:5,000 or 1:10,000), monoclonal mouse anti-human Rpt6 (PSMC5) (Invitrogen, MA, USA, 2SU-1B8, 1:10,000), monoclonal mouse anti-human Rpt4 (Enzo, p42-23, 1:1,000), mouse monoclonal anti-yeast Rpt4 (gift from W. Tansey, 1:2,500), rabbit polyclonal anti-Rpt5 (Enzo), rabbit polyclonal anti-Pre6 (gift from D. Wolf, 1:5,000) and anti-Rpb1 (RNA Pol II) monoclonal mouse antibody (Active Motif, CA, USA, 4H8, 1:2,000). For rabbit primary antibodies, the HRP-linked anti-rabbit IgG secondary antibody (GE Healthcare, IL, USA, NA934) was used at a dilution of 1:5,000 or 1:10,000. For mouse primary antibodies, the HRP-linked anti-mouse secondary antibody (GE Healthcare, NXA931V) was used at a dilution of 1:10,000.

Blots were visualized by enhanced chemiluminescence on a G:Box imaging system with GeneSnap software (Syngene, India). Images were processed with ImageJ software.

## Protection of ubiquitylated species in whole cell yeast lysates

Yeast *ubp8Δ* cells expressing Flag-tagged histone H2B [44] were grown in YPD medium and harvested during exponential phase growth. The cell pellet was washed with water, flash-frozen, and lysed by grinding under liquid nitrogen in a mortar. Proteins were extracted by addition of lysis buffer (50 mM Tris•HCl, pH 7.5, 150 mL NaCl, 1 mM EDTA, 10% glycerol, cOmplete EDTA-free protease inhibitor cocktail (Roche, Switzerland), 1 mM PMSF) in the presence of 20 mM NEM, 3 μM OtUBD, 3 μM 4xTR-TUBE (all final concentrations), or nothing. The resulting lysates were cleared by centrifugation at 21,000 x *g* for 12 minutes at 4˚C and incubated at room temperature for 1 to 4 hours. Flag-tagged H2B was purified by anti-Flag immunoprecipitation with ANTI-FLAG M2 Affinity Gel (Millipore) following the manufacturer's protocol. Whole cell lysates were analyzed by anti-ubiquitin immunoblotting. The anti-Flag precipitates were analyzed by anti-Flag immunoblotting.

## Pulldown with MBP-tagged bait proteins

Pulldowns of MBP-tagged fusion proteins were performed using an amylose resin (New England Biolabs). Appropriate amounts of MBP or MBP fusion proteins (see Figs 1D, S1A and S1B) diluted in 300 μL amylose column buffer (20 mM Tris•HCl, pH 7.5, 200 mM NaCl, 1 mM EDTA) were incubated with 50 μL amylose resin for 1 hour at 4˚C with rotation. The resin was pelleted by centrifugation at 5,000 x *g* for 30 seconds, and the supernatant was removed. One mL of yeast lysate (1 to 2 mg/mL) prepared in column buffer freshly supplemented with protease and DUB inhibitors (cOmplete mini EDTA-free (Roche), 10 mM NEM, 1 mM PMSF) was added to the beads. (For detailed methods of lysate preparation, see section

"Ubiquitin pulldown with protein-linked resins" below.) The mixture was incubated with rotation at 4˚C for 2 hours. The resin was washed 5 times with 1 mL column buffer and then eluted by incubating with column buffer containing 50 mg/mL maltose for 2 hours at 4˚C with rotation. Alternatively, bound proteins could be eluted by incubating with SDS sample buffer for 15 minutes at room temperature.

In the alternative incubation method described in S1B Fig, MBP-OtUBD was first incubated with yeast lysate for 4 hours at 4˚C with rotation. The mixture was then added to the amylose resin and incubated with rotation at 4˚C for another 2 hours, followed by the same washing and elution steps described above.

## Generation of covalently linked affinity purification resins

**OtUBD resin.** Covalently linked OtUBD resin was made by conjugating Cys-OtUBD or Cys-His$_6$-OtUBD to SulfoLink coupling resin (Thermo) according to the manufacturer's protocol. Briefly, 2 mL (bed volume) of SulfoLink resin was placed in a gravity column and equilibrated with 4 bed volumes of SulfoLink coupling buffer (50 mM Tris•HCl, 5 mM EDTA, pH 8.5). Four mg of Cys-OtUBD was diluted in 4 mL coupling buffer supplemented with 20 mM TCEP and incubated at room temperature with rotation for 30 minutes. The diluted Cys-OtUBD was loaded onto the SulfoLink resin, and the mixture was incubated at room temperature for 30–60 minutes with rotation. (Cys-His$_6$-OtUBD required a longer incubation time (60 minutes) than Cys-OtUBD (30 minutes).) The resin was allowed to settle for another 30 minutes before being drained and washed once with 6 mL coupling buffer. Moreover, 4 mL freshly prepared 50 mM L-cysteine dissolved in coupling buffer (pH adjusted to 8.5 with NaOH) was added to the resin and the mixture was incubated at room temperature for 30 minutes with rotation. The resin was allowed to settle for another 30 minutes before being drained and washed with 12 mL 1 M NaCl followed by 4 mL OtUBD column buffer (50 mM Tris•HCl, 150 mM NaCl, 1 mM EDTA, 0.5% Triton-X, 10% glycerol, pH 7.5). For long-term storage (more than 2 days), the resin was stored in column buffer containing 0.05% NaN$_3$ and kept at 4˚C. The resin could be stored at 4˚C for at least 1 year without significant loss of efficacy.

**Negative control Cys-coupled resin.** The negative control resin was made by capping the reactive groups of the SulfoLink resin with cysteine following the manufacturer's protocol. Specifically, 2 mL (bed volume) of resin was incubated with 4 mL freshly prepared 50 mM L-cysteine dissolved in coupling buffer (pH 8.5) at room temperature for 30 minutes with rotation. The resin was subsequently treated and stored as described above for the Cys-OtUBD resin.

**TR-TUBE resin.** TR-TUBE resin was made by conjugating Cys-His$_6$-4xTR-TUBE to the SulfoLink resin following similar procedures as the OtUBD resin with some modifications. In particular, 4.52 mg of Cys-His$_6$-4xTR-TUBE was diluted in 2 mL SulfoLink coupling buffer supplemented with 2 M guanidine•HCl and 20 mM TCEP. Guanidine was added to minimize precipitation of 4xTR-TUBE protein during incubation. Two mL of the diluted TUBE solution was added per 1 mL of SulfoLink resin and the mixture was incubated at room temperature for 1 hour with rotation. The rest of the preparation steps were the same as for OtUBD resin.

**Dsk2 resin.** Dsk2 resin was made similarly as the OtUBD resin except that 6.71 mg of Cys-His6-Dsk2 diluted to 2 mL was added per 1 mL of SulfoLink resin.

**TR-UBA resin.** TR-UBA resin was made similarly as the TR-TUBE resin except that 1.5 mg of Cys-His6-TR-UBA diluted to 2 mL was added per 1 mL SulfoLink resin.

**FK2 resin.** FK2 anti-ubiquitin antibodies were covalently linked to a Protein-G resin following a previous protocol with modifications [20]. Briefly, 500 μg of FK2 mouse monoclonal IgG1 antibody (Cayman Chemical, MI, USA) was diluted in 500 μL DPBS. The solution was

added to 250 μL (bed volume) Protein G Sepharose 4 Fast Flow resin (GE Healthcare) pre-washed with DPBS. The mixture was incubated at 4˚C for 2 hours with rotation. The resin was washed twice with 100 mM triethanolamine•HCl (pH 8.3), and the antibody was then crossed-linked to the resin by incubation with 500 μL 50 mM dimethyl pimelimidate (DMP) freshly dissolved in 100 mM triethanolamine•HCl buffer (pH 8.3) for 4 hours at 4˚C with rotation. The reaction was terminated by incubating with 1.5 mL 100 mM Tris•HCl buffer (pH 7.5) for 2 hours at room temperature. Unconjugated antibody was removed from the resin by washing with 500 μL of 100 mM glycine•HCl buffer (pH 2.5). The resin was equilibrated with DPBS and stored at 4˚C before used.

## Ubiquitin-conjugate purifications with protein-linked resins

### Native conditions

**Preparation of yeast lysate.** In most cases, the frozen yeast pellet was lysed by grinding with mortar and pestle in liquid nitrogen. To 1 volume of the resultant yeast powder, 1 volume of cold native lysis buffer (50 mM Tris•HCl, 300 mM NaCl, 1 mM EDTA, 0.5% Triton-X100, with freshly added 20 mM NEM, cOmplete mini EDTA-free protease inhibitor cocktail (Roche) and 1 mM PMSF, pH 7.5) was added to extract proteins. The mixture was vortexed thoroughly and incubated on ice for 10 minutes with intermittent vortexing. The crude extract was centrifuged at 21,000 x $g$ for 12 minutes, and the supernatant was carefully transferred to a clean tube.

Alternatively, yeast could be lysed by glass bead beating. Cell pellets were resuspended in 1 mL cold native lysis buffer containing 10% glycerol and 0.6 mL acid-washed glass beads (Sigma) and lysed in a FastPrep homogenizer (MP Bio, CA, USA) at 4˚C (5.0 m/s, 3×(30 seconds, 1 minute rest on ice), 4 minutes rest on ice, 3×(30 seconds, 1 minute rest on ice)). The resulting mixture was left on ice for 5 more minutes and then centrifuged at 8,000 x $g$ for 5 minutes at 4˚C. The supernatant was transferred to a new tube while 0.5 mL more lysis buffer was added to the beads and pelleted cell debris. The pellet was resuspended and treated as above. The supernatants were combined and centrifuged at 21,000 x $g$ for 12 minutes at 4˚C. The cleared lysate was transferred to a clean tube.

For mammalian cells, the frozen cell pellets were resuspended in cold native lysis buffer and incubated on ice for 30 to 40 minutes with occasional vortexing. After centrifugation at 21,000 x $g$ for 20 minutes, clarified lysates were transferred to a fresh tube.

Protein concentration in the lysates was measured by the BCA assay, and lysates were adjusted to 2 to 4 mg/mL final protein concentration using native lysis buffer.

**Pulldowns.** A suitable amount of resin was either transferred to a disposable gravity column or, for smaller scale experiments, a microcentrifuge tube. Typically, 25 μL of resin (bed volume) was used for each 1 mg of lysate protein. For the proteomics experiments in this study, 0.25 to 2 mL of resin was used for each pulldown sample. (Here, we describe the procedures used for gravity column-based experiments. For adaption to a microcentrifuge-based experiments, the users could pellet the resin at 1,000 x $g$ for 2 minutes before removal of supernatant.) The storage buffer was drained, and the resin was equilibrated with 5 resin volumes of OtUBD column buffer. If the pulldown was performed for LC-MS/MS analysis, the resin was washed with 2 bed volumes of elution buffer (100 mM glycine•HCl, pH 2.5) and then immediately equilibrated by passing 20 bed volumes of column buffer through the resin.

Lysate was added to the equilibrated resin, the column was capped on both ends, and the mixture was incubated at 4˚C for 2.5 hours with rotation. The resin was allowed to settle for 10 minutes, and the unbound solution was drained and collected as the flow-through. The resin was washed by passing 15 column bed volumes of column buffer, 15 volumes of Wash Buffer-

1 (50 mM Tris•HCl, 150 mM NaCl, 0.05% Tween 20, pH 7.5) and 15 volumes of Wash Buffer-2 (50 mM Tris•HCl, 1 M NaCl, pH 7.5) sequentially through the resin.

**Elution.** If the downstream application was only western blotting, the bound proteins could be eluted by incubating the resin with 2 to 3 resin volumes of 1x SDS sample buffer (50 mM Tris•HCl, pH 6.8, 2% SDS, 5% glycerol, 100 mM DTT, 0.005% bromophenol blue) for 15 minutes at room temperature with rotation. In our hands, the TUBE resin could only be efficiently eluted using this method.

If the purified proteins were to be analyzed by LC-MS/MS, 2 resin volumes of pure water were passed through the resin to push off residue buffers. Then, bound proteins were eluted by incubation in 2 resin volumes of elution buffer (100 mM glycine•HCl, pH 2.5) for 5 minutes at 4˚C with rotation. The eluate was collected and immediately neutralized with 0.2 resin volume of 1M Tris•HCl pH 9 buffer. The elution process was repeated to ensure complete elution (the 2 eluates, E1 and E2, were sometimes combined to give eluate E). In some experiments, the first elution step was done with 100 mM glycine•HCl, pH 3.0 and a third elution step, also with the pH 2.5 buffer, was included to ensure complete elution.

Ubiquitin conjugates in the input, flow-through and eluate for each sample were analyzed by anti-ubiquitin western blotting. The volume loaded onto the SDS-PAGE gel for each sample was normalized to reflect a 1:1:1 scaling of input, flow-through and eluate (e.g., if the total volume of the eluate is 1/10 that of the input, we load 1 volume of the input and 0.1 volume of the eluate on the same SDS-PAGE gel) unless otherwise specified. Total protein from the pulldowns was analyzed by SYPRO Ruby staining of the gels following manufacturer's protocol. SYPRO Ruby stained gels were imaged on a Bio-Rad ChemiDoc imager and quantified using ImageJ software.

### Denaturing conditions

**Preparation of lysates.** Yeast or human cells were lysed as described above for the native condition protocol. After the measurement of protein concentration, the lysates were adjusted to up to 12.58 mg/mL protein with native lysis buffer. The lysate was kept on ice for the whole duration until appropriate amounts of solid urea were added directly to the native lysate to reach a final concentration of 8 M (1 g of urea was added per 0.763 mL of lysate; calculations were based on [48]), and the lysate was vortexed and agitated until the urea had fully dissolved. The urea lysate was incubated at 25˚C for 30 minutes, chilled on ice and diluted 1:1 with native lysis buffer (final concentration of urea, 4 M).

Alternatively, yeast lysate was directly extracted in a urea-containing buffer (Fig 4E, D2 condition). This method may help solubilize precipitated ubiquitylated species. Ground-up yeast powders were resuspended directly in urea lysis buffer (50 mM Tris•HCl, 300 mM NaCl, 8 M urea, 1 mM EDTA, 0.5% Triton-X100, 20 mM NEM, cOmplete mini EDTA-free protease inhibitor cocktail (Roche), 1 mM PSMF, pH 7.5) or the yeast cells could be lysed directly in urea lysis buffer by bead-beating.

The concentration of the cleared lysate was determined by BCA assay and the concentration was adjusted to match other samples. The cleared lysate was incubated at 25˚C for 15 minutes, chilled on ice, and diluted 1:1 with native lysis buffer. This method could in theory include insoluble ubiquitylated proteins and may be useful in specific applications.

**Pulldown and elution protocols.** Pulldown procedures were similar to those described above under the native pulldown protocol except that the first wash step was done with column buffer containing 4 M urea. Elution steps are the same as described in the native pulldown protocol.

## M48 DUB treatment of yeast cell lysates

Yeast powder resulting from grinding the BY4741 strain in liquid nitrogen was reconstituted in M48 lysis buffer (50 mM Tris•HCl, 300 mM NaCl, 1 mM EDTA, 0.5% TritonX, 10% glycerol, pH 7.5, supplemented with 7.6 μM pepstatin A, 5 mM aminocaproic acid (ACA), 5 mM benzamidine, 260 μM AEBSF, 1 mM PMSF, and 1 mM DTT), incubated for 10 minutes on ice and clarified by centrifugation at 21,000 x $g$ at 4˚C. Inhibitors of cysteine proteases were avoided to prevent inhibition of the M48 cysteine protease [84]. Protein concentrations in the lysates were determined by BCA assay, and the lysates were adjusted to 2 to 4 mg/mL protein with M48 lysis buffer. M48 DUB was added to the lysate to give a final enzyme concentration of 100 nM. The mixture was incubated at 37˚C with rotation for 1 hour before subjecting to pulldown analysis. In the control samples where M48 was not added, 10 mM NEM and 20 μM MG132 (a proteasome inhibitor) were also included in the lysis buffer.

## IMAC under denaturing conditions

Eluates from the OtUBD pulldowns were denatured by adding a solid denaturant, either urea to a final concentration of 8 M (used for the results shown in Fig 3E and 3F) or guadinine•HCl to a final concentration of 6 M. (Amounts of denaturants were calculated based on [48].) After the denaturant had fully dissolved, the solution was incubated at 25˚C for 30 minutes before applying to a prewashed HisPur Cobalt resin (Thermo Scientific). The mixture was incubated at room temperature with rotation for 1.5 hours, washed with 8 M urea wash buffer (50 mM Tris•HCl, pH 7.5, 8M urea), and eluted twice, each time by boiling for 5 minutes in 2 resin volumes of 500 mM imidazole in 2x SDS sample buffer (100 mM Tris•HCl, pH 6.8, 4% SDS, 10% glycerol, 200 mM DTT, 0.01% bromophenol blue). Samples were resolved by SDS-PAGE and analyzed by anti-ubiquitin immunoblotting and SYPRO Ruby staining. Specifically, samples containing guanidine were first diluted with 3 portions of pure H$_2$O and then carefully mixed with 4x SDS sample buffer before loaded onto an SDS-PAGE gel to avoid precipitation of SDS.

## Proteomics

**Sample preparation.** Frozen samples were dehydrated on a lyophilizer (Labconco, USA). For all samples except for those in the OtUBD/FK2 comparison experiment, the dry content was reconstituted in pure water and subjected to a methanol-chloroform extraction as described earlier [85].

**In solution protein digestion.** Protein pellets were dissolved and denatured in 8M urea, 0.4M ammonium bicarbonate, pH 8. The proteins were reduced by the addition of 1/10 volume of 45mM DTT (Pierce Thermo Scientific, USA, #20290) and incubation at 37˚C for 30 minutes, then alkylated with the addition of 1/20 volume of 200 mM methyl methanethiosulfonate (MMTS, Pierce Thermo Scientific #23011) with incubation in the dark at room temperature for 30 minutes. Using MMTS avoids the potential false positive identification of GG modification arising from iodoacetamide (IAA) alkylation [86]. The urea concentration was adjusted to 2M by the addition of water prior to enzymatic digestion at 37˚C with trypsin (Promega, WI, USA Seq. Grade Mod. Trypsin, # V5113) for 16 hours. Protease:protein ratios were estimated at 1:50. Samples were acidified by the addition of 1/40 volume of 20% trifluoroacetic acid, then desalted using BioPureSPN PROTO 300 C18 columns (The Nest Group, MA, USA, # HMM S18V or # HUM S18V) following the manufacturer's directions with peptides eluted with 0.1% TFA, 80% acetonitrile. Eluted peptides were speedvaced dry and dissolved in MS loading buffer (2% acetonitrile, 0.2% trifluoroacetic acid). A nanodrop measurement (Thermo Scientific, USA, Nanodrop 2000 UV-Vis Spectrophotometer) determined protein concentrations (A260/A280). Each sample was then further diluted with MS loading buffer to 0.08 μg/μl,

with 0.4μg (5μl) injected for most LC-MS/MS analysis, except for the negative control samples, which were diluted to and injected the same volume as the corresponding OtUBD pulldown samples.

**LC-MS/MS on the Thermo Scientific Q Exactive Plus.** LC-MS/MS analysis was performed on a Thermo Scientific Q Exactive Plus equipped with a Waters (MA, USA) nanoAcquity UPLC system utilizing a binary solvent system (A: 100% water, 0.1% formic acid; B: 100% acetonitrile, 0.1% formic acid). Trapping was performed at 5 μl/min, 99.5% Buffer A for 3 minutes using an ACQUITY UPLC M-Class Symmetry C18 Trap Column (100Å, 5 μm, 180 μm × 20 mm, 2G, V/M; Waters, #186007496). Peptides were separated at 37˚C using an ACQUITY UPLC M-Class Peptide BEH C18 Column (130Å, 1.7 μm, 75 μm X 250 mm; Waters, MA, USA, #186007484) and eluted at 300 nl/min with the following gradient: 3% buffer B at initial conditions; 5% B at 2 minutes; 25% B at 140 minutes; 40% B at 165 minutes; 90% B at 170 minutes; 90% B at 180 minutes; return to initial conditions at 182 minutes. MS was acquired in profile mode over the 300 to 1,700 m/z range using 1 microscan, 70,000 resolution, AGC target of 3E6, and a maximum injection time of 45 ms. Data-dependent MS/MS were acquired in centroid mode on the top 20 precursors per MS scan using 1 microscan, 17,500 resolution, AGC target of 1E5, maximum injection time of 100 ms, and an isolation window of 1.7 m/z. Precursors were fragmented by HCD activation with a collision energy of 28%. MS/MS were collected on species with an intensity threshold of 1E4, charge states 2 to 6, and peptide match preferred. Dynamic exclusion was set to 30 seconds.

**Peptide identification.** Data were analyzed using Proteome Discoverer software v2.2 (Thermo Scientific). Data searching is performed using the Mascot algorithm (version 2.6.1) (Matrix Science) against a custom database containing protein sequences for OtUBD as well as the SwissProt database with taxonomy restricted to *S. cerevisiae* (7,907 sequences) or *Homo sapiens* (20,387 sequences). The search parameters included tryptic digestion with up to 2 missed cleavages, 10 ppm precursor mass tolerance, and 0.02 Da fragment mass tolerance, and variable (dynamic) modifications of methionine oxidation; NEM, NEM+water, carbamidomethyl, or methylthio on cysteine; and GG adduct on lysine, protein amino terminus, serine, threonine or cysteine. Normal and decoy database searches were run, with the confidence level set to 95% ($p < 0.05$). Scaffold (version Scaffold_5.0, Proteome Software, Portland, Oregon, USA) was used to validate MS/MS based peptide and protein identifications. Peptide identifications were accepted if they could be established at greater than 95.0% probability by the Scaffold Local FDR algorithm. Protein identifications were accepted if they could be established at greater than 99.0% probability and contained at least 2 identified peptides. Protein and peptide FDR calculated by Scaffold for each experiment were also reported in S2 Data. The mass spectrometry proteomics data have been deposited to the ProteomeXchange Consortium via the PRIDE [87] partner repository with the dataset identifier PXD032294 (yeast results) and PXD032675 (HeLa cell results).

Quantitative analysis was done by Scaffold 5 (Proteome Software) based on normalized total TIC (MS/MS total ion current). Pearson correlation coefficients were calculated using GraphPad Prism 9 software. Volcano plots were generated using GraphPad Prism 9 software. Proteins are selected as a potential E3 substrate if they meet one of the following criteria: (1) Its average quantitative value (normalized total TIC) is at least 1.5 times higher in the WT samples compared to the E3 deletion samples and $p$-value $< 0.05$. (2) It appeared in at least 3 of the 6 technical replicates of the WT samples but not in any of the 6 technical replicates of the E3 deletion samples. Venn diagrams were generated using the free online tool Multiple List Comparator accessible at https://molbiotools.com/listcompare.php.

GO enrichment analysis on specific protein populations was performed using the online Gene Ontology engine [88–90] accessible at http://geneontology.org.

## Supporting information

**S1 Fig. Additional figures of MBP-OtUBD and immobilized OtUBD-resin pulldowns. (A, B)** Ubiquitin pulldowns with different amounts of MBP-OtUBD. In A, the pulldown was performed by first binding MBP-OtUBD to an amylose resin and then incubating the resin with yeast cell lysate. In B, pulldown was performed by incubating the lysate with MBP-OtUBD and then binding the complexes to amylose resin. U, unbound fraction; E, fraction eluted with maltose. The bands seen at the expected molecular mass of MBP-UBD were likely a result of antibody cross-reactivity. **(C)** Anti-ubiquitin blot of MBP-OtUBD pulldowns from HEK293T whole cell lysates. U, unbound fraction; B, bound fraction (eluted with SDS sample buffer). **(D)** SYPRO Ruby protein stain of the eluates from OtUBD-resin in Fig 2B. E1/E2/E3, eluted fractions from serial low pH elutions. MBP, maltose-binding protein.
(TIF)

**S2 Fig. Presence of proteasomal subunits in OtUBD pulldown samples under different conditions. (A)** Western blots of yeast proteasomal subunits in OtUBD pulldown samples. Unmodified yeast proteasomal subunits (Rpt4, Rpt5, and Pre6) bound to the OtUBD resin under native conditions but not following denaturation of the lysate prior to pulldown. N, Native condition; D, Denaturing condition. **(B)** Western blots of human proteasomal subunits in OtUBD pulldown samples. Unmodified human proteasomal subunit Rpt6 and Rpt4 bound strongly to OtUBD resin under native conditions but only weakly following extract denaturation. Modified Rpt6 and Rpt4 (likely ubiquitylated) bound to the OtUBD resin under both native and denaturing conditions. N, Native condition; D, Denaturing condition.
(TIF)

**S3 Fig. Additional figures for OtUBD pulldown proteomics experiment in yeast. (A)** Representative anti-ubiquitin western blot of OtUBD pulldowns (under native conditions) used for proteomics analysis. IN, input; FT, flow-through; E1/E2/E3, eluted fractions from a series of low pH elutions. **(B)** Representative anti-ubiquitin blot of OtUBD pulldown samples following extract denaturation (urea) and used for proteomics analysis. **(C)** Representative SYPRO Ruby protein stain of OtUBD eluates resolved by SDS-PAGE. **(D)** Number of proteins detected in each biological replicate of OtUBD pulldown-MS and negative control. Error bar represents difference among technical replicates. **(E)** Overlay of TIC chromatographs of representative OtUBD pulldowns and negative control samples. The negative controls overall have much less peptide spectra compared to the OtUBD pulldown samples. This figure is generated with Thermo Xcalibur Qual Browser (v3.0.63) using.raw files from the corresponding runs (QEp21-2054_Zhang_A1_Native_UBD_pos, QEp21-2050_zhang_a2_native_ubd_neg, QEp21-2036_zhang_a3_denatured_ubd_pos, QEp21-2032_zhang_a4_denatured_ubd_neg), which have been deposited to the ProteomeXchange Consortium and made available to the public (see the Methods section for details). **(F)** Adjusted number of proteins detected in each biological replicate of OtUBD pulldowns. Only proteins whose TIC value are at least 20 times higher in the OtUBD pulldown samples compared to the corresponding negative control samples are included.
(TIF)

**S4 Fig. Additional figures for OtUBD pulldown proteomics experiment in HeLa cells and yeast. (A, B)** Anti-ubiquitin western blot of OtUBD pulldowns, and FK2 antibody IPs used for proteomics analysis. OtUBD and FK2 used in A and B are both from different batches. IN, input; FT, flow-through; E, pooled eluted fractions; E1/E2, eluted fractions from a series of low pH elutions. **(C)** Quantitation (estimated from total spectral counts) of different ubiquitin linkages under native and denaturing (urea) conditions in the BY4741 yeast ubiquitylome

from this study. Comparison was made with quantitative data published in Xu and colleagues. The numeric values supporting this panel can be found in S3 Data.
(TIF)

**S5 Fig. Quality control data of the quantitative proteomics analysis with OtUBD pull-downs. (A)** Representative anti-ubiquitin blot of OtUBD pulldowns from WT, *bre1△* and *pib1△* yeast lysates used for proteomics analysis. IN, input; FT, flow-through; E, pooled eluted fractions. **(B)** Representative SYPRO Ruby gel showing the total proteins in eluates from OtUBD pulldown from WT, *bre1△*, and *pib1△* yeast lysates. **(C, D)** Pearson correlation coefficients were calculated between each sample in the analyzed groups using normalized total TIC. Because ubiquitin is present in exceptionally high levels compared to all other proteins, it was excluded from the dataset for this analysis. With the exception of one *pib1Δ* sample, correlations between different samples were generally high, as expected if the majority of the ubiquitylome was not affected by deletion of a single E3. The low correlation in the single *pib1Δ* sample was likely due to an error during sample preparation, so the results were excluded from the quantitation. WT, wild-type.
(TIF)

**S6 Fig. Selective MS/MS spectra from OtUBD pulldown samples. (A)** Representative MS/MS spectrum of the Htb2 K111GG peptide. **(B)** Representative MS/MS spectrum of the Htb2 K123GG peptide. In addition to a/b/y ions, we identified multiple peaks from internal fragmentation and dehydration, potentially due to the serine/threonine-rich nature of the sequence. **(C)** Representative MS/MS spectrum of the YMR160W T534GG peptide.
(TIF)

**S1 Data. List of plasmids, yeast strains, and the sequences of UBDs/TUBEs used in this study.** TUBE, tandem ubiquitin-binding entity; UBD, ubiquitin-binding domain.
(XLSX)

**S2 Data. Proteomics data in this study; this includes numerical values for Fig 6B and 6F.**
(XLSX)

**S3 Data. Numerical values for quantitative analyses in Figs 5C, 5D and S4C.**
(XLSX)

## Acknowledgments

We thank Dr. Chin Leng Cheng and Dr. Hong-Yeoul Ryu for providing yeast strains used in the manuscript. We thank Christian Schlieker's lab at Yale University for sharing the M48 DUB expression plasmid, tissue culture space, and equipment; Candice Paulsen's lab for allowing our use of their ChemiDoc instrument; and David Spiegel's lab for sharing their lyophilizer.

## Author Contributions

**Conceptualization:** Mengwen Zhang, Jason M. Berk, Mark Hochstrasser.

**Data curation:** Mengwen Zhang, Adrian B. Mehrtash, Jean Kanyo.

**Formal analysis:** Mengwen Zhang, Jean Kanyo.

**Funding acquisition:** Mark Hochstrasser.

**Methodology:** Mengwen Zhang.

**Supervision:** Mark Hochstrasser.

**Writing – original draft:** Mengwen Zhang, Mark Hochstrasser.

**Writing – review & editing:** Jason M. Berk, Adrian B. Mehrtash, Jean Kanyo.

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
