## [Editor Report · Decision Letter 0]

29 Nov 2021

Dear Dr Zhang, 

Thank you for submitting your manuscript entitled "A versatile new ubiquitin detection and purification tool derived from a bacterial deubiquitylase" for consideration as a Methods and Resources article by PLOS Biology. Please accept my apologies for the delay in getting back to you as we consulted with an academic editor about your submission. 

Your manuscript has now been evaluated by the PLOS Biology editorial staff, as well as by an academic editor with relevant expertise, and I am writing to let you know that we would like to send your submission out for external peer review.

Once your full submission is complete, your paper will undergo a series of checks in preparation for peer review. Once your manuscript has passed the checks it will be sent out for review. To provide the metadata for your submission, please Login to Editorial Manager (https://www.editorialmanager.com/pbiology) within two working days, i.e. by Dec 01 2021 11:59PM.

If your manuscript has been previously reviewed at another journal, PLOS Biology is willing to work with those reviews in order to avoid re-starting the process. Submission of the previous reviews is entirely optional and our ability to use them effectively will depend on the willingness of the previous journal to confirm the content of the reports and share the reviewer identities. Please note that we reserve the right to invite additional reviewers if we consider that additional/independent reviewers are needed, although we aim to avoid this as far as possible. In our experience, working with previous reviews does save time. 

If you would like to send previous reviewer reports to us, please email me at rhodge@plos.org to let me know, including the name of the previous journal and the manuscript ID the study was given, as well as attaching a point-by-point response to reviewers that details how you have or plan to address the reviewers' concerns. 

Given the disruptions resulting from the ongoing COVID-19 pandemic, please expect some delays in the editorial process. We apologise in advance for any inconvenience caused and will do our best to minimize impact as far as possible.

Kind regards,

Richard

Richard Hodge, PhD

Associate Editor, PLOS Biology

rhodge@plos.org

PLOS

---

## [Decision Letter · Decision Letter 1]

10 Jan 2022

Dear Dr Hochstrasser,

Thank you for submitting your manuscript "A versatile new ubiquitin detection and purification tool derived from a bacterial deubiquitylase" for consideration as a Methods and Resources article at PLOS Biology. Please accept my apologies for the delays that you have experienced during the peer review process. Your manuscript has been evaluated by the PLOS Biology editors, an Academic Editor with relevant expertise, and by four independent reviewers.

The reviews are attached below. You will see that the reviewers are generally positive about the OtUBD method, but also raise overlapping concerns with the strength of the benchmarking experiments to compare OtUBD with current methods (such as diGly and TUBE-based methods), as well as comparing OtUBD with single UBA domains. They also note that additional quantitative assessments should be performed to provide more evidence that OtUBD offers a significant advantage over other methods, such as quantifying the affinity of OtUBD for the seven different ubiquitin linkages. In addition, the reviewers raise overlapping concerns with the number of biological replicates in several of the figure panels and ask that replicate data be provided.

In light of the reviews, we will not be able to accept the current version of the manuscript, but we would welcome re-submission of a much-revised version that takes into account the reviewers' comments. We cannot make any decision about publication until we have seen the revised manuscript and your response to the reviewers' comments. Your revised manuscript is also likely to be sent for further evaluation by the reviewers.

We expect to receive your revised manuscript within 3 months. We do understand that the current circumstances are somewhat difficult, so please do get in touch with us if you need more time. 

**IMPORTANT - SUBMITTING YOUR REVISION**

*Re-submission Checklist*

*Published Peer Review*

*PLOS Data Policy*

*Blot and Gel Data Policy*

Sincerely,

Richard

Richard Hodge, PhD

Associate Editor, PLOS Biology

rhodge@plos.org

REVIEWS:

Reviewer #1: This manuscript extends earlier work from the Hochstrasser lab by developing their previously described high-affinity ubiquitin (Ub) binding domain protein, OtUBD, into a convenient and effective reagent that can be used to capture (poly)Ub species from cell lysates or other complex mixtures. In particular, a covalently-linked resin-bound form OtUBD was shown to have several properties that make it very well-suited for applications in proteomics. Chief among these are the ability of OtUBD to capture monoUb protein conjugates as well as polyUb conjugates, and that either native or denaturing conditions can be used for binding and washing steps during affinity purifications. Although the use of OtUBD as an affinity reagent was perhaps obvious as a follow-up to the original Berk et al. 2020 paper, the methods and findings in this new paper by Zhang et al. are very useful; they are certain to be adopted widely by the very large number of researchers with interests that involve Ub biochemistry.

Overall, this study was performed rigorously and the conclusions are supported by the experimental results. However, one exception (see item #1) and a few other issues are noted below.

1. (p. 7 & Fig. 1D) The authors' concluded from the results in Fig. 1D that, relative to a single OtUBD, a 3xOtUBD tandem fusion protein binds Ub conjugates more efficiently. They speculate that avidity was likely responsible for the difference. However, that difference was seen only when the binding proteins were used at a density of 4.5 nmol/mL resin and disappeared when that was doubled (in fact, the MBP-OtUBD resin now looks like it bound more conjugates than the MBP-3xOtUBD form). A more likely explanation for the difference between the single and 3x OtUBD is that the latter resin had 3-fold more binding domain — i.e., the 4.5 nmol/mL resin with MBP-3xOtUBD had 13.5 nmol/mL OtUBD. The lanes that compared the 9 nmol/mL resins show that even doubling the amount of MBP-OtUBD dramatically increased conjugate binding. Avidity cannot be invoked to explain the greater binding of free monoUb, a monovalent ligand, by 3xOtUBD. Examination of the free Ub bands in the blot suggests that the amount of Ub in the lysate exceeded the capacity of the OtUBD — but not 3xOtUBD — when used at 4.5 nmol/mL, and that doubling the OtUBD allowed almost all of the Ub to be captured. 

2. Pulldown of Ub species by OtUBD appears to capture polyUb conjugates with all types of Ub-Ub linkages, and quantification of the different linkages is in good agreement with an earlier report by Xu et al. 2009 [Note: Fig S4E should note the proper reference on the figure and in the legend]. This is important and useful information. However, the result could be interpreted by readers as suggesting that OtUBD will bind well to a Ub that has any one of its lysine(s) linked by another Ub. That was not shown; in fact, it is possible the some Ub-Ub linkages may interfere with OtUBD binding to a ubiquitinated (i.e., proximal) Ub in a polyUb chain. Nonetheless, because every polyUb chain would contain a distal Ub subject to tight binding by OtUBD, all Ub-Ub linkages would be recovered in the pulldown. It would be helpful if the authors would discuss this point in their paper and elaborate upon what's known and not known regarding linkage specificity of OtUBD.

3. A very important consideration in the use of the OtUBD resin for pulldowns, and especially in the context of unbiased proteomics analyses, is that care must be taken to ensure that the OtUBD binder is used in excess over the total Ub. This point should be made in the paper.

4. (Fig. 3C,F) In panels C and F, it would be helpful to identify the positions expected for free monoUb. It looks like free monoUb was captured efficiently using native conditions but not with denaturing conditions. Is that the case? If so, could that be because there is less functional OtUBD available in the denaturing condition? It would be informative to compare the two binding conditions with respect to the capacity of the OtUBD resin for monoUb.

Some minor points:

5. The structure of the 3xOtUBD (i.e., sequences of the linkers between the OtUBDs) should be included in the Methods or Supplementary Data.

6. The Introduction includes too much review of the ubiquitin system.

Reviewer #2: The manuscript by Zhang, et al., presents a method for purification of ubiquitylated proteins from total yeast and human cell extracts, suitable for combining with mass spectrometry approaches for identification of ubiquitylated proteins and their sites of modification. This is a variation of methods that have been widely developed and employed over at least the last dozen years and is based on the use of isolated ubiquitin-binding domains (UBDs) from various naturally occuring proteins. The innovation here is that the UBD used here (OtUBD) is from a bacterial effector protein with a particularly high affinity for ubiquitin (Kd of approximately 5 nM), as reported recently by the same group in Nature Communications (2020). Some of the advantages claimed for this modified method are that 1) it is more efficient that other methods, 2) that it recognizes all seven ubiquitin linkages and 3) that it recognizes non-canonical (non-lysine-based) ubiquitin linkages. All of these claims require further substantiation and the first two of these require quantitative analyses, which are currently lacking. 

The relevant statement on PLOS Biology standards for methods articles are that, "… the method presented may demonstrate substantial improvements to currently used methodologies. They would need to significantly outperform their predecessors by precision, resolution, speed, accessibility, and/or cost". With this in mind, specific comments follow.

The claim that it the OtUBD matrix recognizes all types of ubiquitin chain linkages is based on identifying these in mass spec experiments using total yeast extracts. This is a good start, but there should be some quantitative assessment of the relative affinity for different linkages types, for example using purified chains of somewhat uniform length. These are of course readily available. As a side note, an experiment using human cell extract did not identify one of the seven linkages, but then it is noted that the experiment was only done with a single replicate. I am wondering why any mass spec experiment done with a single replicate is included in the manuscript, at all. 

Related to the prior point, given that the entire premise of the method is that it is more efficient than other related methods (i.e., using different UBDs), some direct quantitative comparisons to a handful of the differing UBDs in use is in order. While there are some experiments here that compare to the well-known TUBE matrix, even here it is entirely non-quantitative. 

The claim that the method can detect non-canonical ubiquitin conjugates (e.g., ser/thr/cys linkages is not sufficiently discussed or tested. At the very least this needs to be presented more completely and carefully. 

In summary, the potential advance here is technical and not conceptual. As such, it is important to include additional quantitative assessments to substantiate the central claims. 

Reviewer #3: Zhang et al. "A versatile new ubiquitin detection and purification tool derived from a bacterial deubiquitylase"

In this paper, Zhang and colleagues characterize the functionality of OtUBD, an isolated ubiquitin binding domain from a bacterial deubiquitylating enzyme, as a ubiquitin enrichment reagent. This work extends the findings of a manuscript published last year by the same group which included extensive biochemical analyses of OtUBD, and a crystal structure of OtUBD bound to ubiquitin. Despite extensive successful efforts by the field to characterize the ubiquitylated proteome, our ability to reliably detect specific ubiquitylated proteins remains limited. In large part, this inability is due to the imperfect reagents (di-gly antibodies, TUBEs, epitope-tagged ubiquitin) currently available to enrich the ubiquitylated proteome from lysates. Here the authors demonstrate the ability of recombinant OtUBD to isolate ubiquitylated species from yeast and human cell lysates. They show that OtUBD protects ubiquitylated proteins from deubiquitylation after cell lysis. Additionally, the authors use OtuBD to purify ubiquitylated species under both native and denaturing conditions. Finally, they use OtUBD to monitor ubiquitylation on specific proteins, as well as globally survey the ubiquitylated proteome via mass spectrometry.

Zhang et al. nicely demonstrate the utility of OtUBD in ubiquitin enrichment. Specifically, the ability of OtUBD to bind monoubiquitylated substrates represents a significant advantage over the currently used TUBE-based reagents, which, as the authors note, do not have a high affinity for monoubiquitin. Thus, I think with some improvements, this study can have broad interest in the field who may want to try out an OtuBD-based enrichment strategy to study the ubiquitylation of, or by, a protein of interest. 

Major comments:

1. The authors do a fairly rigorous characterization of the OtUBD. However, they mainly compare its efficacy to a tandem ubiquitin binding entity (commercially available which I think is in an inappropriate comparison. The authors themselves make a tandem reagent with the OtUBD and find it less effective than the single domain reagent. Thus, the better and more appropriate comparison is to a reagent with a single UBA (the yeast DSK2-UBA is a commonly used ubiquitin affinity reagent, but there is a large list of potential single domain ubiquitin affinity regents that can be used (UIM, UBD, UBA, etc). Even if these are not commercially available, the authors certainly have the ability to produce one of these regents in a similar manner as the OtUBD. This is important for two reasons. First, the authors state the higher affinity of OtUBD as one of the characteristics that likely make it a better ubiquitin capture reagent. As such, comparison to the yeast DSK2-UBA (or other single domain UBA affinity reagent) may reveal the OtUBD to bind much more ubiquitinated proteins and be a better reagent (compared to the current comparison with the TUBE). Second the authors make the specific point (in figure 1 and figure 4) that the OtUBD is better at binding (via protection in figure 1) mono-Ub proteins than the TUBE. This is expected given the previous studies showing that a single ubiquitin binding domain captures more mono-ubiquitinated proteins than tandem domains. Here the comparison to a single UBA (like the DSK2-UBA) is a more appropriate comparison. I recommend repeating some of these studies using this more appropriate comparison. Similarly, in figure 4B, the authors compare OtUBD to TUBE enrichments under native conditions but don't have a similar comparison under denaturing conditions where they point out a specific enrichment for a putative mono-Ub modified Rpt5. Having a TUBE-based and a single-UBA-based enrichment for comparison would be useful to evaluate these results.

2. In figure 1D, at the lower concentration of bait protein, the 3xUBD enriches ubiquitylated proteins more efficiently than the single UBD. However, the reverse is true with the higher concentration of bait protein. Why is this the case?

3. Because this is a methods development study, comparison to the current standards is important. I appreciate the proteomics characterization of proteins enriched using OtUBD as an affinity handle. The authors compare the results they obtained from yeast lysates to three other studies using di-Gly-based proteomics methods. It would also be useful to compare the results to other proteomics experiments using TUBE-based or other ubiquitin-binding domain affinity methods as that is the most similar to the OtUBD method investigated here. Similarly, when the authors performed a similar proteomic study using mammalian cell lysates, the authors compare their results to those obtained from a previous study using FK2-antibody-based enrichments from a study published in 2005. It would be useful to have some uniformity to the comparisons that are being done. I suggest the authors compare their mammalian proteomics results with di-GLY-based studies (similar to what they did with the yeast proteomics results, and there have been many such mammalian di-GLY proteomics studies in the last 10 years). Again, also comparing these results with recent mammalian-based TUBE or ubiquitin binding proteomics results, such as those reported for TR-TUBE itself in Yoshida et al 2015, or Lear et al. J Biol Chem (2020) would be the best comparison to the OtUBD results reported here. Making relevant comparison to more recent studies will be essential in evaluating any methodological gains from using the OtUBD-based enrichment strategy reported in this study.

4. While OtUBD binds the high molecular weight ubiquitin "smears", it shows a clear preference for binding free ubiquitin in lysates. This is seen, for example, in the UBD FT lanes in Fig 2C, where free ubiquitin is depleted more completely than the high molecular weight species. If the unbound fractions were subsequently incubated with additional OtUBD, could a more complete enrichment be achieved? If not, it will be important to understand the nature of the ubiquitylated proteins that are not well captured by OtUBD.

5. In supplemental figure 4E, the authors report total TIC as a box-and-whisker plot in which the relevant controls are not interpretable. It would be more useful to just show the overlaid total TIC chromatograph from each of the runs. As it is currently reported, I am not sure what the box or whiskers represent. 

6. In the supplemental table reporting the proteomic results, in table 2 reporting peptides with di-GLY modifications, there are some examples where the modification is listed on the terminal amino acid (A6ZYV5 for example with the ILENDLk peptide). This suggests that the data has been inappropriately processed as trypsin does not cut at ub-modified lysine residues. I suggest the authors re-process their data to not allow C-term di-GLY modifications as these are almost certainly false-positive identifications. 

Minor comments:

1. In line 285, the figure reference should be for 3A, not 4A.

2. In line 330, the figure reference should be for S4G, not D.

3. In line 337, the figure reference should be for S4H, not E.

4. In line 339, there is no reference given for the HeLa proteomics data.

Reviewer #4: SUMMARY

In this manuscript, Zhang et al. sought to establish a novel tool to detect and purify ubiquitin by using a high-affinity ubiquitin-binding domain derived from a bacterial deubiquitylase from Orientia tsutsugamushi. For this purpose, the authors characterized the ubiquitylome and ubiquitin-associated proteome of yeast and compared ubiquitylated substrates from yeast and human samples to assess the efficacy of existing methods. In an initial screen of ubiquitylation protection and pulldown experiments, the authors show that OtUBD is capable of protecting ubiquitylated proteins from DUBs and of enriching these targets. The authors were able to generate a covalently-linked OtUBD resin for ubiquitylated protein purification and by introducing the V203D mutation in OtUBD, they further determined its binding affinity and specificity. Using the OtUBD pulldown in denaturing versus native conditions revealed the enrichment of proteins that are covalently bound to ubiquitin. Moreover, the authors could demonstrate consistent results with OtUBD for human samples. In further pulldown experiments under distinct conditions and by using yeast strains which lack the ubiquitin E3 ligase Bre1 or the DUB Ubp8, the authors found that OtUBD resin can facilitate the detection of both, monoubiquitylated and polyubiquitylated proteins. The authors could verify this result in human cells. After establishing the method, the authors used OtUBD pulldown proteomics on native and denatured lysates from yeast and human cells. Interestingly, the authors found much more enriched ubiquitylated proteins in comparison to results obtained from immunoprecipitation using FK2. Lastly, the authors showed that OtUBD and label-free quantification facilitates the identification of potential E3 substrates which was challening in the past due to the instability and low abundance of ubiquitylated proteins and the transient nature of E3-substrate interaction. Moreover, this approach could have great potential for the identification of substrates of other ubiquitin-related enzymes. Overall, the study of Zhang and colleagues clearly expands the methodological toolbox for exploring and analysing monoubiquitylated and polyubiquitylated substrates. There are only a few minor issues that the authors should consider. 

1) Is there any explanation why there is more binding of ubiquitylated substrates with a higher bait protein concentration in the MBP-UBD than in the MBP-3xUBD as it is the opposite for a lower concentration?

2) Is there any deeper meaning of the striking band in the flowthrough of the native sample in the anti-ubiquitin blot of fractions from Co2+ IMAC?

3) The authors should perform at least three biological replicates for their analysis. Especially for their work in HeLa cells. Currently, they seemed to have done only one experiment in this setting.

4) Please make sure that the labeling in the figures is clear and consistent (e.g. under regard of upper- and lower case).

5) Please make sure that every IB panel carries a complete molecular weight marker labeling.

---

## [Decision Letter · Decision Letter 2]

10 May 2022

Dear Dr Hochstrasser,

Thank you for submitting your revised Methods and Resources article entitled "A versatile new ubiquitin detection and purification tool derived from a bacterial deubiquitylase" for publication in PLOS Biology. I have now obtained advice from the original reviewers and have discussed their comments with the Academic Editor.

As you can see, the reviewers appreciated the additional data included in the revised manuscript to address their comments. Based on the reviews, I am pleased to say that we will probably accept this manuscript for publication, provided you address the remaining concerns outlined by Reviewer #3. In addition, please make sure to address the following data and other policy-related requests that I have provided below (points A-F):

(A) Since the tool enables the detection of ubiquitylated proteins, we would like to propose the following two title suggestions, to make it more compelling for our broad readership. 

" A versatile new tool derived from a bacterial deubiquitylase to detect and purify ubiquitylated substrates and their interacting proteins"

OR

“A versatile new tool derived from a bacterial deubiquitylase to detect and purify ubiquitylated substrates”

(B) You may be aware of the PLOS Data Policy, which requires that all data be made available without restriction: http://journals.plos.org/plosbiology/s/data-availability. For more information, please also see this editorial: http://dx.doi.org/10.1371/journal.pbio.1001797

Regardless of the method selected, please ensure that you provide the individual numerical values that underlie the summary data displayed in the following figures, as they are essential for readers to assess your analysis and to reproduce it.

Figure 5C-D, 6B, 6F, S3E, S4C

(C) Please ensure that the data deposited in the ProteomeXchange database (accession numbers PXD032294 and PXD032675) are made publicly available at this stage.

(D) We require the original, uncropped and minimally adjusted images supporting all blot and gel results reported in the following figures:

Figure 1C-D, 2B-F, 3C-I, 4A-G, S1A-D, S2A-B, S3A-D, S3F, S4A-B, S5A-B

We will require these files before a manuscript can be accepted so please prepare and upload them now. Please carefully read our guidelines for how to prepare and upload this data: https://journals.plos.org/plosbiology/s/figures#loc-blot-and-gel-reporting-requirements

(E) Please also ensure that each of the relevant figure legends in your manuscript include information on *WHERE THE UNDERLYING DATA CAN BE FOUND*, and ensure your supplemental data file/s has a legend.

(F) Please ensure that your Data Statement in the submission system accurately describes where your data can be found and is in final format, as it will be published as written there. This includes referring to underlying data provided in the Supplementary Information and removing the sentence ‘Login and passwords for reviewers are provided in the cover letter’

We expect to receive your revised manuscript within two weeks.

*Published Peer Review History*

*Early Version*

Sincerely,

Richard

Richard Hodge, PhD

Associate Editor, PLOS Biology

rhodge@plos.org

REVIEWS:

Reviewer #1: In their revised manuscript, the authors have thoroughly addressed all my concerns. I have no doubt that the results from this high-quality study will be used extensively by researchers in the ubiquitin field.

Reviewer #2: The authors have responded throughly to all of the reviewers' comments and have included new data and clarifications where needed. This work will represent a significant contribution to the ubiquitin field and I anticipate that the methods described here will be rapidly and widely adopted by researchers in the field. 

Reviewer #3: The revised manuscript has addressed the majority of our concerns. We would request the author clear up a few outstanding issues prior to publication (we do not need to see another revision). 

1. We appreciate adding the direct comparison of OtUBD to the full-length DSK2 (it was a little odd that the authors used full length over the isolated UBA and claimed the full-length worked better given the full DSK2 protein will have a Ub fold at the N-term which can interact in trans with the UBA domain likely impairing ub binding in the lysate). In new figure 4, under native conditions, OtUBD and Dsk2 seem to isolate different ubiquitylated Rpb1 species (the smear vs the triplet bands in Figure 4F). It is beyond the scope of this study to identify the various species, but we would request that the authors acknowledge and comment on this difference in the main text.

2. It would be useful to see a silver stain or coomassie stain from the comparative pull down experiments to show that equal amounts of the OtUBD, DSK2, TR-UBA, TR-TUBE are present in the pulldowns. This will help validate the usefulness of the cross comparison (what if there is 4x less DSK2 on resin).

3. Last, we have some concerns about how the mass spec data, particularly how the ub-modified peptides were analyzed. The authors say they manually removed all identifications with C-term GG modifications. While this is useful in that those are certainly incorrect identifications, this will also alter the false discovery rate for the modified peptides. The presence of those false positives in dataset indicate that many other false positives are present in the dataset that cannot be simply identified and removed as they did for the c-term modified peptides. In fact, it is entirely unclear how the modified peptides were identified and processed as the methods simply state that "GG modified peptides were further analyzed using

Scaffold PTM 3.3 software". The authors also suggest that Mascot is incapable of filtered search outputs with C-term modifications. There are at least two publicly available MS processing suites (Maxquant being one of them) that are capable of processing these modified peptides correctly. It very unclear how the authors controlled the false-discovery rate for K-modified, or even more importantly, T or S di-gly modified peptides. When calculating the false-discovery rate, the number of incorrect matches (usually reversed peptides) that also have the K-GG or S-GG need to be considered separately from the unmodified peptides. How many reverse matches have S-GG or T-GG? How does that compare to the forward matches reported in their tables? My guess is that the calculated FDR, when considering modified peptides separately, will be MUCH higher than is reported here. This is very important when reporting these peptides because if their identification is equally likely to be a false positive, there is no value in reporting those identifications (in fact that are as likely to be wrong as they are right). Mascot and certainly Maxquant will be able to control FDR for modified peptides only. 

Reviewer #4: The authors adequately resolved all of the critical points that I raised - either by revising the manuscript or by performing additional experiments that corroborated their initial data. Therefore, I am happy to recommend this study for publication. Well done!

---

## [Editor Report · Decision Letter 3]

30 May 2022

Dear Dr Hochstrasser,

Thank you for the submission of your revised Methods and Resources article "A versatile new tool derived from a bacterial deubiquitylase to detect and purify ubiquitylated substrates and their interacting proteins" for publication in PLOS Biology. On behalf of my colleagues and the Academic Editor, Kylie Walters, I am pleased to say that we can accept your manuscript for publication, provided you address any remaining formatting and reporting issues. These will be detailed in an email you should receive within 2-3 business days from our colleagues in the journal operations team; no action is required from you until then. Please note that we will not be able to formally accept your manuscript and schedule it for publication until you have completed any requested changes.

PRESS

Sincerely, 

Richard

Richard Hodge, PhD

Associate Editor, PLOS Biology

rhodge@plos.org

PLOS
